# Optimal Sample Complexity of $M$-wise Data for Top-$K$ Ranking

**Minje Jang**[*]
School of Electrical Engineering
KAIST
jmj427@kaist.ac.kr

**Sunghyun Kim**[*]
Electronics and Telecommunications Research Institute
Daejeon, Korea
koishkim@etri.re.kr

**Changho Suh**
School of Electrical Engineering
KAIST
chsuh@kaist.ac.kr

**Sewoong Oh**
Industrial and Enterprise Systems Engineering Department
UIUC
swoh@illinois.edu

## Abstract

We explore the top-$K$ rank aggregation problem in which one aims to recover a consistent ordering that focuses on top-$K$ ranked items based on partially revealed preference information. We examine an $M$-wise comparison model that builds on the Plackett-Luce (PL) model where for each sample, $M$ items are ranked according to their perceived utilities modeled as noisy observations of their underlying true utilities. As our result, we characterize the minimax optimality on the sample size for top-$K$ ranking. The optimal sample size turns out to be inversely proportional to $M$. We devise an algorithm that effectively converts $M$-wise samples into pairwise ones and employs a spectral method using the refined data. In demonstrating its optimality, we develop a novel technique for deriving tight $\ell_\infty$ estimation error bounds, which is key to accurately analyzing the performance of top-$K$ ranking algorithms, but has been challenging. Recent work relied on an additional maximum-likelihood estimation (MLE) stage merged with a spectral method to attain good estimates in $\ell_\infty$ error to achieve the limit for the pairwise model. In contrast, although it is valid in slightly restricted regimes, our result demonstrates a spectral method alone to be sufficient for the general $M$-wise model. We run numerical experiments using synthetic data and confirm that the optimal sample size decreases at the rate of $1/M$. Moreover, running our algorithm on real-world data, we find that its applicability extends to settings that may not fit the PL model.

## 1 Introduction

Rank aggregation has been explored in a variety of contexts such as social choice [15, 6], web search and information retrieval [20], recommendation systems [7], and crowd sourcing [16], to name a few. It aims to bring a consistent ordering to a collection of items, given partial preference information.

---

[*]Equal contribution.

Preference information can take various forms depending on the context. One such form, which we examine in this paper, is ordinal; preferences for alternatives are represented as an ordering. Consider crowd-sourced data collected by annotators asked to rank a few given alternatives based on their preference. The aggregated data can be used to identify the most preferred. One example can be a review process for conference papers (e.g., NIPS) where reviewers are asked to not only review papers, but also order them based on how much they enjoy them. The collected data could be used to highlight papers that may interest a large audience. Alternatively, consider sports (races or the like) and online games where a number of players compete. One may wish to rank them according to skill.

Its broad range of applications has led to a volume of work done. Of numerous schemes developed, arguably most dominant paradigms are spectral algorithms [14, 20, 37, 41, 47, 45] and maximum likelihood estimation (MLE) [22, 28]. Postulating the existence of underlying real-valued preferences of items, they aim to produce preference estimates consistent in a global sense, e.g., measured by low squared loss. But such estimates do not necessarily guarantee optimal ranking accuracy. Accurate ranking has more to do with how well the ordering of estimates matches that of the true preferences, and less to do with how close the estimates are to the true preferences minimizing overall errors. Moreover, in practice, what we expect from accurate ranking is an ordering that precisely separates only a few items ranked highest from the rest, not an ordering that respects the entire items.

**Main contributions.** In light of it, we explore top-$K$ ranking which aims to recover the correct set of top-ranked items only. We examine the Plackett-Luce (PL) model which has been extensively explored [24, 18, 5, 25, 38, 43, 33, 4]. It is a special case of random utility models [46] where true utilities of items are presumed and a user's revealed preference is a partial ordering according to noisy manifestations of the utilities. It satisfies the 'independence of irrelevant alternatives' property in social choice theory [34, 35] and is the most popular model in studying human choice behavior given multiple alternatives (see Section 2). It is well-known that it subsumes as a special case the Bradley-Terry-Luce (BTL) model [12, 32] which concerns two items. We consider an $M$-wise comparison model where comparisons are given as a preference ordering of $M$ items. In this setting, we characterize the minimax limit on the sample size (i.e., sample complexity) needed to reliably identify the *set* of top-$K$ ranked items, which turns out to be inversely proportional to $M$. To the best of our knowledge, it is the first result that characterizes the limit under an $M$-wise comparison model.

In achieving the limit, we propose an algorithm that consists of sample breaking and *Rank Centrality* [37], one spectral method we choose among other variants [10, 9, 37, 33]. First, it converts $M$-wise samples into many more pairwise ones, and in doing so, it carefully chooses only $M$ out of all $\binom{M}{2}$ pairwise samples obtainable from each $M$-wise sample. This sample breaking (see Section 3.1) extracts only the essential information needed to achieve the limit from given $M$-wise data. Next, using the refined pairwise data, the algorithm runs a spectral method to identify top-ranked items.

A novel technique we develop to attain tight $\ell_\infty$ estimation error bounds has been instrumental to our progress. Analyzing $\ell_\infty$ error bounds is a critical step to characterizing the minimax sample complexity for top-$K$ ranking as presented in [17], but has been technically challenging. Even after decades of research since the introduction of spectral methods and MLE, two dominant approaches in the field, we lack notable results for tight $\ell_\infty$ error bounds. This is largely because techniques proven useful to obtain good $\ell_2$ error bounds do not translate into attaining good $\ell_\infty$ error bounds. In this regard, our result contributes to progress on $\ell_\infty$ error analysis (see Section 3.2 and the supplementary).

We can compare our result to that of [17] by considering $M = 2$. Although the two optimal sample complexities match, the conditions under which they do differ; our result turns out to be valid under a slightly restricted condition (see Section 3.3). In terms of achievability, the algorithm in [17] merges an additional MLE stage with a spectral method, whereas we employ only a spectral method. From numerical experiments, we speculate that the condition under which the result of [17] holds may not be sufficient for spectral methods alone to achieve optimality (see Section 4.1).

We conduct numerical experiments to support our result. Using synthetic data, we show that the minimax optimal sample size indeed decreases at the rate of $1/M$. We run our algorithm on real-world data collected from a popular online game (*League of Legends*) and find its applicability to extend to settings that may not necessarily match the PL model. From the collected data, we extract $M$-wise comparisons and rank top users in terms of skill. We examine its robustness aspect against partial data and also evaluate its rank result with respect to the official rank *League of Legends* provides. In both cases, we compare it with a counting-based algorithm [42, 11] and demonstrate its advantages.

**Related work.** To the best of our knowledge, [17] investigated top-$K$ identification under the random comparison model of interest for the first time. A key distinction here is that we examine the random *listwise* comparison model based on the PL model. *Rank Centrality* was developed in [37] based on which we devise our ranking scheme tailored for listwise comparison data.

In the PL model, some viewed ranking as parameter estimation. Maystre and Grossglauser [33] developed an algorithm that shares a spirit of spectral ranking and showed its performance is the same as MLE for estimating underlying preference scores. Hajek *et al.* [25] derived minimax lower bounds of parameter estimation error, and examined gaps with upper bounds of MLE as well as MLE with a rank-breaking scheme that decomposes partial rankings into pairwise comparisons.

Some works examined several sample breaking methods that convert listwise data into pairwise data in the PL model. Azari Soufiani *et al.* [5] considered various methods to see if they sustain some statistical property in parameter estimation. It examined full breaking that converts an $M$-wise sample into $\binom{M}{2}$ pairwise ones, and adjacent breaking that converts an ordinal $M$-wise sample into $M-1$ pairwise ones whose associated items are adjacent in the sample. Ashish and Oh [4] considered a method that converts an $M$-wise sample into multiple pairwise ones and assigns different importance weights to each, and examined the method on several types of comparison graphs.

There are a number of works that explored ranking problems in different models and with different interests. Some works [43, 2] have adopted PAC (probably approximately correct) [44] or regret [21, 8, 23] as their metric to allow some margin of error, in contrast to our work where 0/1 loss (the most stringent criterion) is considered to investigate the worst-case scenario (see Section 2). Rajkumar and Agarwal [40] put forth statistical assumptions that ensure the convergence of rank aggregation methods including *Rank Centrality* and MLE to an optimal ranking. Active ranking where samples are obtained adaptively has received attention as well. Jamieson and Nowak [29] considered perfect total ranking and characterized the query complexity gain of adaptive sampling in the noise-free case, and the works of [29, 1] explored the query complexity in the presence of noise aiming at approximate total rankings. Recently, Braverman *et al.* [13] considered three noisy models, examining if their algorithm can achieve reliable top-$K$ ranking. Heckel *et al.* [27] considered a model where noisy pairwise observations are given, with a goal to partition the items into sets of pre-specified sizes based on their scores, which includes top-$K$ ranking as a special case. Mohajer *et al.* [36] considered a fairly general noisy model which subsumes as special cases various models. They derived upper bounds on the sample size required for reliable top-$K$ sorting as well as top-$K$ partitioning, and showed that active ranking can provide significant gains over passive ranking.

## 2 Problem Formulation

**Notation.** We denote by $[n]$ to represent $\{1, 2, \ldots, n\}$, and by $\mathcal{G} = ([n], \mathcal{E}^{(M)})$ to represent an $M$-wise comparison graph in which total $n$ vertices reside and each hyper-edge is connected if there is a comparison among $M$ vertices, and $d_i$ to represent the out-degree of vertex $i$.

**Comparison model and assumptions.** Suppose we perform a few evaluations on $n$ items. We assume the comparison outcomes are generated based on the PL model [39]. We consider $M$-wise models where the comparison outcomes are obtained in the form of a preference ordering of $M$ items.

*Preference scores.* The PL model assumes the existence of underlying preferences $\boldsymbol{w} := \{w_1, w_2, \ldots, w_n\}$, where $w_i$ represents the preference score of item $i$. The outcome of each comparison depends solely on the latent scores of the items being compared. Without loss of generality, we assume that $w_1 \geq w_2 \geq \cdots \geq w_n > 0$. We assume the range of scores to be fixed irrespective of $n$. For some positive constants $w_{\min}$ and $w_{\max}$, $w_i \in [w_{\min}, w_{\max}], 1 \leq i \leq n$. We note that the case where the range $w_{\max}/w_{\min}$ grows with $n$ can be translated into the above fixed-range regime by separating out those items with vanishing scores (e.g. via a voting method like Borda count [11, 3]).

*Comparison model.* We denote by $\mathcal{G} = ([n], \mathcal{E}^{(M)})$ a comparison graph where a set of $M$ items $\mathcal{I} = \{i_1, i_2, \ldots, i_M\}$ are compared if and only if $\mathcal{I}$ belongs to the hyper-edge set $\mathcal{E}^{(M)}$. We examine random graphs, constructed in a similar manner according to the Erdős-Rényi random graph model; each set of $M$ vertices is connected by a hyper-edge independently with probability $p$. Notice that when $M = 2$, such random graphs we consider follow precisely the Erdős-Rényi random model.

*M*-wise comparisons. We observe $L$ samples for each $\mathcal{I} = \{i_1, i_2, \ldots, i_M\} \in \mathcal{E}^{(M)}$. Each sample is an ordering of $M$ items in order of preference. The outcome of the $\ell^{\text{th}}$ sample, denoted by $s_{\mathcal{I}}^{(\ell)}$, is generated according to the PL model: $s_{\mathcal{I}}^{(\ell)} = (i_1, i_2, \ldots, i_M)$ with probability $\prod_{m=1}^{M} \left( w_{i_m} / \sum_{r=m}^{M} w_{i_r} \right)$, where item $i_a$ is preferred over item $i_b$ in $\mathcal{I}$ if $i_a$ appears to the left of $i_b$, which we also denote by $i_a \succ i_b$. We assume that conditional on $\mathcal{G}$, $s_{\mathcal{I}}^{(\ell)}$'s are jointly independent over $\mathcal{I}$ and $\ell$. We denote the collection of all samples by $\boldsymbol{s} := \{s_{\mathcal{I}} : \mathcal{I} \in \mathcal{E}^{(M)}\}$, where $s_{\mathcal{I}} = \{s_{\mathcal{I}}^{(1)}, s_{\mathcal{I}}^{(2)}, \ldots, s_{\mathcal{I}}^{(L)}\}$.

**Performance metric and goal.** Given comparison data, one wishes to know whether or not the top-$K$ ranked items are identifiable. We consider the probability of error $P_e$ in identifying the correct *set* of the top-$K$ ranked items: $P_e(\psi) := \mathbb{P}\{\psi(\boldsymbol{s}) \neq [K]\}$, where $\psi$ is any ranking scheme that returns a set of $K$ indices and $[K]$ is the set of the first $K$ indices. Our goal in this work is to characterize the *admissible region* $\mathcal{R}_{\boldsymbol{w}}$ of $(p, L)$ in which top-$K$ ranking is feasible for a given PL parameter $\boldsymbol{w}$, in other words, $P_e$ can be vanishingly small as $n$ grows. The admissible region $\mathcal{R}_{\boldsymbol{w}}$ is defined as $\mathcal{R}_{\boldsymbol{w}} := \{(p, L) : \lim_{n \to \infty} P_e(\psi(\boldsymbol{s})) = 0\}$. In particular, we are interested in the minimax *sample complexity* of an estimator defined as $S_\delta := \inf_{p \in [0,1], L \in \mathbb{Z}^+} \sup_{\boldsymbol{v} \in \Omega_\delta} \{\binom{n}{M} pL : (p, L) \in \mathcal{R}_{\boldsymbol{v}}\}$, where $\Omega_\delta = \{\boldsymbol{v} \in \mathbb{R}^n : (v_K - v_{K+1})/v_{\max} \geq \delta\}$. Note that this definition shows that we conservatively examine minimax scenarios where nature behaves adversely with the worst-case $\boldsymbol{w}$.

## 3 Main Results

Separating the two items near the decision boundary (i.e., the $K^{\text{th}}$ and $(K+1)^{\text{th}}$ ranked items) is key in top-$K$ ranking. Unless the gap is large enough, noise in the observations leads to erroneous estimates which no ranking scheme can overcome. We pinpoint a separation measure as $\Delta_K := (w_K - w_{K+1})/w_{\max}$, which turns out to be crucial in establishing the fundamental limit.

Noted in [22], if a comparison graph $\mathcal{G}$ is not connected, it is impossible to determine the relative preferences between two disconnected entities. Thus, we assume all comparison graphs to be connected. To guarantee it, for a hyper-random graph with edge size $M$, we assume $p > \log n / \binom{n-1}{M-1}^2$.

Now, let us formally state our main results. First, for comparison graphs under $M$-wise observations, we establish a *necessary* condition for top-$K$ ranking.

**Theorem 1.** *Fix $\epsilon \in (0, \frac{1}{2})$. Given an $M$-wise comparison graph $\mathcal{G} = ([n], \mathcal{E}^{(M)})$, if*
$$\binom{n}{M} pL \leq c_0(1 - \epsilon) \frac{n \log n}{\Delta_K^2} \frac{1}{M}, \tag{1}$$
*for some numerical constant $c_0$, then for any ranking scheme $\psi$, there exists a preference score vector $\boldsymbol{w}$ with separation measure $\Delta_K$ such that $P_e(\psi) \geq \epsilon$.*

The proof is a generalization of Theorem 2 in [17], and we provide it in the supplementary. Next, for comparison graphs under $M$-wise observations, we establish a *sufficient* condition for top-$K$ ranking.

**Theorem 2.** *Given an $M$-wise comparison graph $\mathcal{G} = ([n], \mathcal{E}^{(M)})$ and $p \geq c_1(M-1)\sqrt{\frac{\log n}{\binom{n-1}{M-1}}}$, if*
$$\binom{n}{M} pL \geq c_2 \frac{n \log n}{\Delta_K^2} \frac{1}{M}, \tag{2}$$
*for some numerical constants $c_1$ and $c_2$, then* Rank Centrality *correctly identifies the top-$K$ ranked items with probability at least $1 - 2n^{-\frac{1}{15}}$.*

We provide the proof of Theorem 2 in the supplementary. From below, we describe the algorithm we use, sample breaking and *Rank Centrality* [37], and soon give an outline of the proof.

Note that Theorem 1 gives a necessary condition of the sample complexity $S_{\Delta_K} \gtrsim n \log n / M\Delta_K^2$ and Theorem 2 gives a sufficient condition of it $S_{\Delta_K} \lesssim n \log n / M\Delta_K^2$, and they match. That is, we establish the minimax optimality of *Rank Centrality*: $n \log n / M\Delta_K^2$.

---

$^2 p > \log n / \binom{n}{M-1}$ is derived in [19] as a sharp threshold for connectivity of hyper-graphs. We assume a slightly more strict condition for ease of analysis. This does not make a big difference in our result, as the two conditions are almost identical order-wise given $M < n/2$, a reasonable condition for regimes where $n$ is large.

## 3.1 Algorithm description

---

**Algorithm 1** Rank Centrality [37]

---

**Input** the collection of statistics $\boldsymbol{s} = \left\{ s_\mathcal{I} : \mathcal{I} \in \mathcal{E}^{(M)} \right\}$.
**Convert** the $M$-wise sample for each hyper-edge $\mathcal{I}$ into $M$ pairwise samples:
      1. Choose a circular permutation of the items in $\mathcal{I}$ uniformly at random,
      2. Break it into the $M$ pairs of adjacent items, and denote the set of pairs by $\phi(\mathcal{I})$,
      3. Use the (pairwise) data of the pairs in $\phi(\mathcal{I})$.

**Compute** the transition matrix $\hat{\boldsymbol{P}} = [\hat{P}_{ij}]_{1 \le i,j \le n}$: $\hat{P}_{ij} = \begin{cases} \frac{1}{2d_{\max}} y_{ij} & \text{if } i \ne j; \\ 1 - \sum_{k:k\ne j} \hat{P}_{kj} & \text{if } i = j; \\ 0 & \text{otherwise.,} \end{cases}$

where $d_{\max}$ is the maximum out-degree of vertices in $\mathcal{E}^{(M)}$.
**Output** the stationary distribution of matrix $\hat{\boldsymbol{P}}$.

---

*Rank Centrality* aims to estimate rankings from pairwise comparison data. Thus, to make use of $M$-wise comparison data for *Rank Centrality*, we apply a sample breaking method that converts $M$-wise data into pairwise data. To be more specific, if there is a hyper-edge $\mathcal{I} = \{1, 2, \ldots, M\}$, we choose a circular permutation of the items in $\mathcal{I}$ uniformly *at random*. Suppose we pick a circular permutation $(1, 2, \ldots, M-1, M, 1)$. Then, we break it into $M$ pairs of items in the order specified by the permutation: $\{1,2\}, \{2,3\}, \ldots, \{M-1, M\}, \{M, 1\}$ (see Section 3.3 for a remark on why we do not lose optimality by our sample breaking method). Let us denote by $\phi(\mathcal{I})$ this set of pairs. We use the converted pairwise comparison data associated with the pairs in $\phi(\mathcal{I})$[3]:

$$y_{ij,\mathcal{I}}^{(\ell)} = \begin{cases} 1 & \text{if } \{i,j\} \in \phi(\mathcal{I}) \text{ and } i \succ j; \\ 0 & \text{otherwise} \end{cases}, \quad y_{ij} := \sum_{\mathcal{I}:\{i,j\} \in \phi(\mathcal{I})} \frac{1}{L} \sum_{\ell=1}^{L} y_{ij,\mathcal{I}}^{(\ell)}. \quad (3)$$

In an ideal scenario where we obtain an infinite number of samples per $M$-wise comparison, i.e., $L \to \infty$, sufficient statistics $\frac{1}{L} \sum_{\ell=1}^{L} y_{ij,\mathcal{I}}^{(\ell)}$ converge to $w_i/(w_i + w_j)$. Then, the constructed matrix $\hat{\boldsymbol{P}}$ defined in Algorithm 1 becomes a matrix $\boldsymbol{P}$ whose entries $[P_{ij}]_{1 \le i,j \le n}$ are defined as

$$P_{ij} = \begin{cases} \frac{1}{2d_{\max}} \sum_{\mathcal{I}:\{i,j\} \in \phi(\mathcal{I})} \frac{w_i}{w_i + w_j} & \text{for } \mathcal{I} \in \mathcal{E}^{(M)}; \\ 1 - \sum_{k:k \ne j} P_{kj} & \text{if } i = j; \\ 0 & \text{otherwise.} \end{cases} \quad (4)$$

The entries for observed item pairs represent the relative likelihood of item $i$ being preferred over item $j$. Intuitively, random walks of $\boldsymbol{P}$ in the long run visit some states more often, if they have been preferred over other frequently-visited states and/or preferred over many other states. The random walks are reversible as $w_i P_{ji} = w_j P_{ij}$ holds, and irreducible under the connectivity assumption. Once we obtain the unique stationary distribution, it is equal to $\boldsymbol{w} = \{w_1, \ldots, w_n\}$ up to some constant scaling. It is clear that random walks of $\hat{\boldsymbol{P}}$, a noisy version of $\boldsymbol{P}$, will give us an approximation of $\boldsymbol{w}$.

## 3.2 Proof outline

We outline the proof of Theorem 2 by introducing Theorem 3, which we show leads to Theorem 2.

**Theorem 3.** *When* Rank Centrality *is employed, with high probability, the $\ell_\infty$ norm estimation error is upper-bounded by*

$$\frac{\|\hat{\boldsymbol{w}} - \boldsymbol{w}\|_\infty}{\|\boldsymbol{w}\|_\infty} \lesssim \sqrt{\frac{n \log n}{\binom{n}{M} pL}} \sqrt{\frac{1}{M}}, \quad (5)$$

*where $p \ge c_1(M-1)\sqrt{\frac{\log n}{\binom{n-1}{M-1}}}$, and $c_1$ is some numerical constant.*

Let $\|\boldsymbol{w}\|_\infty = w_{\max} = 1$ for ease of presentation. Suppose $\Delta_K = w_K - w_{K+1} \gtrsim \sqrt{\log n / \binom{n}{M} pL} \sqrt{1/M}$. Then, $\hat{w}_i - \hat{w}_j \geq w_i - w_j - |\hat{w}_i - w_i| - |\hat{w}_j - w_j| \geq w_K - w_{K+1} - 2\|\hat{\boldsymbol{w}} - \boldsymbol{w}\|_\infty > 0$, for all $1 \leq i \leq K$ and $j \geq K+1$. That is, the top-$K$ items are identified as desired. Hence, as long as $\Delta_K \gtrsim \sqrt{\log n / \binom{n}{M} pL} \sqrt{1/M}$, i.e., $\binom{n}{M} pL \gtrsim n \log n / M \Delta_K^2$, reliable top-$K$ ranking is achieved with the sample size of $n \log n / M \Delta_K^2$.

Now, let us prove Theorem 3. To find an $\ell_\infty$ error bound, we first derive an upper bound on the point-wise error between the score estimate of item $i$ and its true score, which consists of three terms:

$$|\hat{w}_i - w_i| \leq |\hat{w}_i - w_i| \hat{P}_{ii} + \sum_{j:j\neq i} |\hat{w}_j - w_j| \hat{P}_{ij} + \left| \sum_{j:j\neq i} (w_i + w_j) \left( \hat{P}_{ji} - P_{ji} \right) \right|. \tag{6}$$

We can obtain (6) from $\hat{\boldsymbol{w}} = \hat{\boldsymbol{P}} \hat{\boldsymbol{w}}$ and $\boldsymbol{w} = \boldsymbol{P} \boldsymbol{w}$. We then obtain upper bounds on the three terms:

$$\hat{P}_{ii} < 1, \left| \sum_{j:j\neq i} (w_i + w_j) \left( \hat{P}_{ji} - P_{ji} \right) \right| \lesssim \sqrt{\frac{n \log n}{\binom{n}{M} pL}} \sqrt{\frac{1}{M}}, \sum_{j:j\neq i} |\hat{w}_j - w_j| \hat{P}_{ij} \lesssim \sqrt{\frac{n \log n}{\binom{n}{M} pL}} \sqrt{\frac{1}{M}}, \tag{7}$$

with high probability (Lemmas 1, 2 and 3 in the supplementary). (7) ends the proof. We obtain the first two from Hoeffding's inequality. The last is key; this is where we sharply link an $\ell_2$ error bound of $\sqrt{n \log n / \binom{n}{M} pL} \sqrt{1/M}$ (Theorem 4 in the supplementary) to the desired $\ell_\infty$ error bound (5).

On the left hand side of the third inequality, the point-wise error of item $j$ which affects that of item $i$ as expressed in (6), may not be captured for some $j$, since there may be no hyper-edge that includes items $i$ and $j$. This makes it hard to draw a link from the obtained $\ell_2$ error bound to the inequality, since $\ell_2$ errors can be seen as the sum of all point-wise errors. To include them all, we recursively apply (6) to $|\hat{w}_j - w_j|$ in the third inequality and then apply the rest two properly (for detailed derivation, see the beginning of the proof of Lemma 3 in the supplementary). Then, we get

$$\sum_{j:j\neq i} |\hat{w}_j - w_j| \hat{P}_{ij} \lesssim \sum_{j:j\neq i} \sum_{k:k\neq j} |\hat{w}_k - w_k| \hat{P}_{jk} \hat{P}_{ij} + \sqrt{\frac{n \log n}{\binom{n}{M} pL}} \sqrt{\frac{1}{M}}. \tag{8}$$

Manipulating the first term of the right hand side (for derivation, see the proof of Lemma 3), we get

$$\sum_{k=1}^n |\hat{w}_k - w_k| \sum_{j:j\notin\{i,k\}} \hat{P}_{jk} \hat{P}_{ij} \leq \|\hat{\boldsymbol{w}} - \boldsymbol{w}\|_2 \sqrt{\sum_{k=1}^n \left( \sum_{j:j\notin\{i,k\}} \hat{P}_{jk} \hat{P}_{ij} \right)^2}. \tag{9}$$

We show that $\sum_{j:j\notin\{i,k\}} \hat{P}_{jk} \hat{P}_{ij}$ concentrates on the order of $1/n$ for all $k$'s in the proof of Lemma 3. Since $\|\boldsymbol{w}\|_2 \leq \sqrt{n} \|\boldsymbol{w}\|_\infty = \sqrt{n}$, we get $\|\hat{\boldsymbol{w}} - \boldsymbol{w}\|_2 / \sqrt{n} \leq \|\hat{\boldsymbol{w}} - \boldsymbol{w}\|_2 / \|\boldsymbol{w}\|_2$. We derive this $\ell_2$ error bound to be $\sqrt{n \log n / \binom{n}{M} pL} \sqrt{1/M}$ (Theorem 4 in the supplementary), matching (5).

To describe the concentration of $\sum_{j:j\notin\{i,k\}} \hat{P}_{jk} \hat{P}_{ij}$, we need to consider dependencies in it. To see them, we upper-bound it as follows (for details, see the proof of Lemma 3 in the supplementary).

$$\sum_{j:j\notin\{i,k\}} \hat{P}_{ij} \hat{P}_{jk} \leq \frac{1}{4d_{\max}^2} \sum_{j:j\notin\{i,k\}} \sum_{\mathcal{I}_1:i,j\in\mathcal{I}_1, \mathcal{I}_2:j,k\in\mathcal{I}_2} X_{\mathcal{I}_1 \mathcal{I}_2}, \tag{10}$$

where $X_{\mathcal{I}_1 \mathcal{I}_2} := \mathbb{I}\left[ \{i,j\} \in \phi(\mathcal{I}_1) \right] \mathbb{I}\left[ \{j,k\} \in \phi(\mathcal{I}_2) \right]$. For $M > 2$, there can exist $j_a$ and $j_b$ such that $\{i, j_a, j_b\} \in \mathcal{I}_1$, $j_a \in \mathcal{I}_2$ and $j_b \notin \mathcal{I}_2$. Then, summing over $j$, $X_{\mathcal{I}_1 \mathcal{I}_2}$ and $X_{\mathcal{I}_1 \mathcal{I}_3}$, where $\mathcal{I}_3$ is another hyper-edge that includes $j_b$ and $k$, are dependent concerning the same hyper-edge $\mathcal{I}_1$. To handle this, we use Janson's inquality [30], one of concentration inequalities that consider dependencies.

To derive a necessary condition matching our sufficient condition, we use a generalized version of Fano's inequality [26] as in the proof of Theorem 3 in [17] and complete combinatorial calculations.

### 3.3 Discussion

*Optimality versus $M$ — intuition behind our sample breaking method:* For each $M$-wise sample, we form a circular permutation uniformly at random, and extract $M$ pairwise samples each of which concerns two adjacent items in it. Suppose we have an $M$-wise sample $1 \prec 2 \prec \cdots \prec M$, and for simplicity we happen to form a circular permutation as $(1, 2, \ldots, M-1, M, 1)$; we extract $M$ pairwise samples as $1 \prec 2, 2 \prec 3, \ldots, (M-1) \prec M, 1 \prec M$. Let us provide the intuition behind why this leads us to the optimal sample complexity. For the case of $M = 2$, *Rank Centrality* achieves the optimal order-wise sample complexity of $n \log n / \Delta_K^2$ as characterized in [17]. In addition, one $M$-wise sample in the PL model can be broken into $M - 1$ *independent* pairwise ones, since pairwise data of two arbitrary items among the $M$ items depend on the true scores of the two items only. In our example, one can convert the $M$-wise sample into $M - 1$ independent pairwise ones as $1 \prec 2, 2 \prec 3$, $\ldots, (M-1) \prec M$. From these, it is intuitive to see that we can achieve reliable top-$K$ ranking with an order-wise sample complexity of $n \log n / (M-1)\Delta_K^2$ by converting each $M$-wise sample into $M - 1$ independent pairwise ones. Notice a close gap to the optimal sample complexity in Section 3.

*Tight $\ell_\infty$ error bounds:* As shown in Section 3.2, deriving a tight $\ell_\infty$ error bound is critical to analyzing the performance of a top-$K$ ranking algorithm. Recent work [17] has relied on combining an additional stage of local refinement in series with *Rank Centrality* to derive it, and characterized the optimal sample complexity for the pairwise model. In contrast, although it is valid in a slightly restricted regime (see the next remark), we employ only *Rank Centrality* and still succeed in achieving optimality for the $M$-wise model that includes the pairwise model. Deriving tight $\ell_\infty$ error bounds being crucial, it is hard for one to attain this result without a fine analytical technique. It is our main theoretical contribution to develop one. For details, see the proof of Lemma 3 in the supplementary that sharply links an $\ell_\infty$ error bound (Theorem 3 therein) and an $\ell_2$ error bound (Theorem 4 therein). *Rank Centrality* has been shown to achieve the performance nearly as good as MLE in terms of $\ell_2$ error, but little has been known in terms of $\ell_\infty$ error, until now. Our result has made clear progress.

*Analytical technique:* Our analysis is not limited to *Rank Centrality*. Whenever one wishes to compute the difference between the leading eigenvector of any matrix and that of its noisy version, one can obtain (6), (8) and (9). Thus, it can be adopted to link $\ell_2$ and $\ell_\infty$ error bounds for any spectral method.

*Dense regimes:* Our main result concerns a slightly denser regime, indicated by the condition $p \gtrsim (M-1)\sqrt{\log n / \binom{n-1}{M-1}}$, where many distinct groups of items are likely to be compared. One can see that this dense regime condition is not necessary for top-$K$ ranking; for the pairwise case $M = 2$, it is $p \gtrsim \log n / n$ as shown in [17]. However, it is not clear yet whether or not the dense regime condition is required under our approach that employs only a spectral method. Our speculation from numerical experiments is that the sparse regime condition, $\log n / \binom{n-1}{M-1} \lesssim p \lesssim (M-1)\sqrt{\log n / \binom{n-1}{M-1}}$, may not be sufficient for spectral methods to achieve reliable top-$K$ ranking (see Section 4).

## 4 Experimental Results

### 4.1 Synthetic data simulation

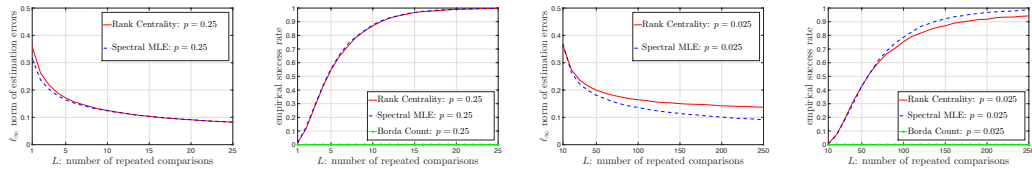

Figure 1: Dense regime ($p_{\text{dense}} = 0.25$, first two figures): empirical $\ell_\infty$ estimation error v.s. $L$ (left); empirical success rate v.s. $L$ (right). Sparse regime ($p_{\text{sparse}} = 0.025$, last two figures): empirical $\ell_\infty$ estimation error v.s. $L$ (left); empirical success rate v.s. $L$ (right).

First, we conduct a synthetic data experiment for $M = 2$, the pairwise comparison model, to compare our result in Theorem 2 to that in recent work [17]. We consider both dense ($p \gtrsim \sqrt{\log n / n}$) and sparse ($\log n / n \lesssim p \lesssim \sqrt{\log n / n}$) regimes. We set constant $c_1 = 2$, and set $p_{\text{dense}} = 0.25$ and $p_{\text{sparse}} = 0.025$, to make each be in its proper range. We use $n = 500$, $K = 10$, and $\Delta_K = 0.1$. Each result in all numerical simulations is obtained by averaging over 10000 Monte Carlo trials.

In Figure 1, the first two figures show the experiments in the dense regime. We see that as $L$ increases, meaning as we obtain pairwise samples beyond the minimal sample complexity, (1) the $\ell_\infty$ error of *Rank Centrality* decreases and meets that of *Spectral MLE* (left); (2) the success rate of *Rank Centrality* increases and soon hits $100\%$ along with *Spectral MLE* (right). The curves support our result; in the dense regime $p \gtrsim \sqrt{\log n/n}$, *Rank Centrality* alone can achieve reliable top-$K$ ranking. The last two figures show the experiments in the sparse regime. We see that as $L$ increases, (1) the $\ell_\infty$ error of *Rank Centrality* decreases but does not meet that of *Spectral MLE* (left); (2) the success rate of *Rank Centrality* increases but does not reach that of *Spectral MLE* which hits nearly $100\%$ (right). The curves lead us to speculate that the sparse regime condition $\log n/n \lesssim p \lesssim \sqrt{\log n/n}$ may not be sufficient for spectral methods to achieve reliable top-$K$ ranking.

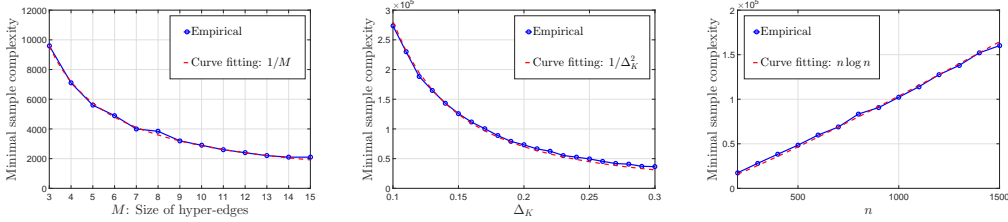

Figure 2: Empirical minimal sample complexity v.s. $M$ (first), $\Delta_K$ (second), and $n \log n$ (third).

Next, we corroborate our optimal sample complexity result in Theorem 2. We examine whether the empirical minimal sample complexity decreases at the rate of $1/M$ and $1/\Delta_K^2$, and increases at the rate of $n \log n$. To verify its reduction at the rate of $1/M$, we run experiments for $M$ ranging from 3 to 15. We increase the number of samples by increasing $p$ until the success rate reaches $95\%$ for each $M$. The number of samples we use to achieve it is considered as the empirical minimal sample complexity for each $M$. We set the other parameters as $n = 100$, $L = 20$, $K = 5$ and $\Delta_K = 0.3$. The result for each $M$ in all simulations is obtained by averaging over 1000 Monte Carlo trials. To verify the other two relations, we follow similar procedures. As for $1/\Delta_K^2$, we set $n = 200$, $M = 2$, $L = 20$ and $K = 5$. As for $n \log n$, we set $M = 2$, $L = 4$, $K = 5$ and $\Delta_K = 0.4$.

The first figure in Figure 2 shows the reduction in empirical minimal sample complexity with a blue solid curve. The red dashed curve is obtained by curve-fitting. We can see that the empirical minimal sample complexity drops inversely proportional to $M$. From the second and third figures, we can see that in terms of $\Delta_K$ and $n \log n$, it also behaves as our result in Theorem 2 predicts.

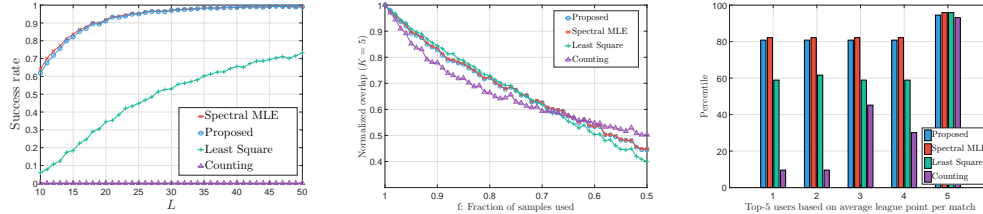

Figure 3: (First) Empirical success rates of four algorithms: our algorithm (blue circle), heuristic *Spectral MLE* (red cross), least square (green plus), and counting (purple triangle); (Second) Top-5 ranked users: normalized overlap v.s. fraction of samples used; (Third) Top-5 users' (sorted by average *League of Legends* points earned per match) percentile in the ranks by our algorithm, heuristic *Spectral MLE*, least square, and counting. For instance, the user who earns largest points per match (first entry) is at around the 80-th percentile according to our algorithm and heuristic *Spectral MLE*, the 60-th percentile according to least square, and the 10-th percentile according to counting.

Last, we evaluate the success rates of various algorithms on $M$-wise comparison data. We consider our proposed algorithm, *Spectral MLE*, least square (*HodgeRank* [31]), and counting. Since *Spectral MLE* has been developed for pairwise data, we heuristically extend it. We apply our sample breaking method to obtain pairwise data needed. For any parameters required to run *Spectral MLE*, we heuristically find the best ones which give rise to the highest success rate. In the other two algorithms, we first apply our sample breaking method as well. Then, for least square, we find a score vector $\hat{w}$ such that the squared error $\sum_{(i,j) \in \mathcal{E}} \left(\log(\hat{w}_i/\hat{w}_j) - \log(y_{ij}/y_{ji})\right)^2$, where $\mathcal{E}$ is the edge set for the converted pairwise data, is minimized. For counting, we count each item's number of wins in all

involved pairwise data. We use $n = 100$, $M = 4$, $p = 0.0025 \cdot (M-1)\sqrt{\log n / \binom{n-1}{M-1}}$, $K = 5$ and $\Delta_K = 0.3$. Each result in all simulations is obtained by averaging over 5000 Monte Carlo trials.

The first figure in Figure 3 shows that our algorithm and heuristic *Spectral MLE* perform best (the latter being marginally better), achieving near-100% success rates for large $L$. It also shows that they outperform the other two algorithms which do not achieve near-100% success rates even for large $L$.

## 4.2  Real-world data simulation

One natural setting where we can obtain $M$-wise comparison data is an online game. Users randomly get together and play, and the results depend on their skills. We find *League of Legends* to be a proper fit[4]. In extracting $M$-wise data, we adopt a measure widely accepted as a factor that rates users' skill in the user community[5]. We incorporate this measure into our model as follows. For each match ($M$-wise sample), we have 10 users, each associated with its measure. In breaking $M$-wise samples, for each user pair $(i, j)$, we compare their measures and declare user $i$ wins if its measure is larger than user $j$'s. This corresponds to $y_{ij}^{(\ell)}$ in our model. We assign 1 if user $i$ wins and 0 otherwise. They may play together in multiple, say $L_{ij}$, matches. We can compute $y_{ij} := (\sum_{\ell=1}^{L_{ij}} y_{ij}^{(\ell)})/L_{ij}$ to use for *Rank Centrality*. As $M$-wise data is extracted from team competitions, *League of Legends* does not perfectly fit our model. Yet one main reason to run this experiment is to see whether our algorithm works well in other settings that do not necessarily fit the PL model, being broadly applicable.

We first investigate the robustness aspect by evaluating the performance against *partial* information. To this end, we use all collected data and obtain a ranking result for each algorithm which we consider as its baseline. Then, for each algorithm, we reduce sample sizes by discarding some of the data, and compare the results to the baseline to see how robust each algorithm is against partial information. We conduct this experiment for four algorithms: our proposed algorithm, the heuristic extension of *Spectral MLE*, least square and counting.

We choose our metric as a normalized overlap: $|\mathcal{S}_{\mathsf{comp}} \cap \mathcal{S}_{\mathsf{part}}|/K$, where $K = 5$, $\mathcal{S}_{\mathsf{comp}}$ is the set of top-$K$ users identified using the complete dataset and $\mathcal{S}_{\mathsf{part}}$ is that identified using partial datasets. In choosing partial data, we set $f \in (0.5, 1)$, and discard each match result with probability $f$ independently. We compute the metric for each $f$ by averaging over 1000 Monte Carlo trials.

The second figure of Figure 3 shows that over the range of $f$ where overlaps above 60% are retained, our algorithm, along with some others, demonstrates good robustness against partial information.

In addition, we compare the ranks estimated by the four algorithms to the rank provided by *League of Legends*. By computing the average points earned per match for each user, we infer the rank of the users determined by official standards. In the third figure of Figure 3, the $x$-axis indicates the top-5 users identified by computing average *League of Legends* points earned per match and sorting them in descending order. The $y$-axis indicates the percentile of these top-5 users according to the ranks by the algorithms of interest. Notice that the top-5 ranked users by *League of Legends* standards are also placed at high ranks when ranked by our algorithm and heuristic *Spectral MLE*; they are all placed at the 80-th percentile or above. On the other hand, most of them (4 users out of the top-5 users) are placed at noticeably lower ranks when ranked by least square and counting.

## 5  Conclusion

We characterized the minimax (order-wise) optimal sample complexity for top-$K$ rank aggregation in the $M$-wise comparison model that builds on the PL model. We corroborated our result using synthetic data experiments and verified the applicability of our algorithm on real-world data.

## Acknowledgments

This work was supported by Institute for Information & communications Technology Promotion(IITP) grant funded by the Korea government(MSIT) (2017-0-00694, Coding for High-Speed Distributed Networks).

## Footnotes

[3]In comparison, the adjacent breaking method [5] directly follows the ordering evaluated in each sample; if it is $1 \prec 2 \prec \cdots \prec M-1 \prec M$, it is broken into pairs of adjacent items: $1 \prec 2$ up to $M-1 \prec M$. Our method turns out to be consistent, i.e., $\frac{\Pr[y_{ij}=1]}{\Pr[y_{ji}=0]} = \frac{w_i}{w_j}$ (see (4)), whereas the adjacent breaking method is not [5].

[4]Two teams of 5 users compete. Each user kills an opponent, assists a mate to kill one, and dies from an attack. At the end, one team wins, and different points are given to the users. We use users' kill/assist/death data (non-negative integers), which can be considered as noisy measurements of their skill, and rank them by skill.

[5]We define a measure as $\{(\# \text{ of kills} + \# \text{ of assists})/(1 + \# \text{ of deaths})\} \times \mathsf{weight}$. We adopt this measure since it is similar to the one officially provided (called KDA statistics). We assign winning users a weight of 1.1 and losing users a weight of 1.0, to give extra credit (10%) to users who lead their team's winning.

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
