[Supplementary Material]

<div align="center">

# Supplementary Material
# Proofs of Theorem 1 and Theorem 2

</div>

## Organization of Proofs

We first prove Theorem 2 in Section 1, which gives us a sufficient condition for top-$K$ ranking. We then prove Theorem 1 in Section 4, which gives us a necessary condition for top-$K$ ranking.

To prove Theorem 2, we establish two additional theorems. One theorem (Theorem 3 in Section 2) describes an $\ell_\infty$ estimation error bound, and the other (Theorem 4 in Section 3) describes an $\ell_2$ estimation error bound. To begin with, we show that Theorem 3 implies Theorem 2, and make the proof of Theorem 2 boil down to proving Theorem 3.

The proof of Theorem 3 is, as emphasized in the paper, where we make our theoretical contribution; we sharply link the $\ell_2$ error bound (derived in Theorem 4) to the $\ell_\infty$ error bound, leading them to be on the *same* order. Lemma 3 plays a key role in establishing the link. It applies Janson's inequality (Janson 2002) stated in Lemma 4 to describe the concentration behavior of sums of *dependent* random variables.

In the proof of Theorem 4, we derive the $\ell_2$ error bound that we use in proving Theorem 3. It adopts a similar line of steps to that taken in (Negahban, Oh and Shah 2016). The key difference is that we perform more involved calculations, as we consider a more general model.

To prove Theorem 1, we make use of the generalized Fano's inequality due to (Han and Verdú 1994). The line of steps we follow is similar to that taken in the proof of Theorem 2 in (Chen and Suh 2015). However, the details of the steps are more involved including combinatorial calculations, as we consider a more general model.

We note that, to help enhance readability for proofs that involve many steps and sub-proofs, we provide a brief outline at the beginning of each proof.

## 1 Proof of Theorem 2

**Theorem 2.** *Given an $M$-wise comparison graph $\mathcal{G} = ([n], \mathcal{E}^{(M)})$ and $p \geq c_1(M-1)\sqrt{\frac{\log n}{\binom{n-1}{M-1}}}$, if*

$$\binom{n}{M}pL \gtrsim \frac{n \log n}{\Delta_K^2}\frac{1}{M}, \tag{1}$$

*then* Rank Centrality *correctly identifies the top-$K$ ranked items with high probability, where $c_1$ is some numerical constant.*

**Proof:** To distinguish the top-$K$ items from the rest, the pointwise error of each item becomes a fundamental bottleneck for top-$K$ ranking. It will be impossible to separate the $K^{th}$ and $(K+1)^{th}$ ranked items unless their score separation exceeds the aggregate error of the score estimates for the two items. Based on this observation, we figure out an upper bound of the maximal pointwise error $\|\hat{\boldsymbol{w}} - \boldsymbol{w}\|_\infty$, where $\boldsymbol{w}$ is the ground-truth preference score vector and $\hat{\boldsymbol{w}}$ is an estimate of $\boldsymbol{w}$, in Theorem 3. We will soon show that Theorem 3 implies Theorem 2.

**Theorem 3.** $\ell_\infty$ *norm estimation errors can be upper-bounded by*

$$\frac{\|\hat{\boldsymbol{w}} - \boldsymbol{w}\|_\infty}{\|\boldsymbol{w}\|_\infty} \lesssim \sqrt{\frac{n \log n}{\binom{n}{M}pL}}\sqrt{\frac{1}{M}}, \tag{2}$$

<div align="center">

</div>

*with high probability, where $p \geq c_1(M-1)\sqrt{\frac{\log n}{\binom{n-1}{M-1}}}$, and $c_1$ is some numerical constant.*

Let us first show that Theorem 3 implies Theorem 2. Later, we will provide the proof of Theorem 3 in detail. Let $\|\boldsymbol{w}\|_\infty = w_{\max} = 1$ for ease of representation. Suppose $\Delta_K = w_K - w_{K+1} \gtrsim \sqrt{\frac{\log n}{\binom{n}{M}pL}}\sqrt{\frac{1}{M}}$, then

$$\hat{w}_i - \hat{w}_j \geq w_i - w_j - |\hat{w}_i - w_i| - |\hat{w}_j - w_j| \geq w_K - w_{K+1} - 2\|\hat{\boldsymbol{w}} - \boldsymbol{w}\|_\infty > 0, \tag{3}$$

for all $1 \leq i \leq K$ and $j \geq K+1$, indicating that the algorithm outputs the top-$K$ items as desired. Hence, as long as $\Delta_K \gtrsim \sqrt{\frac{\log n}{\binom{n}{M}pL}}\sqrt{\frac{1}{M}}$, in other words, $\binom{n}{M}pL \gtrsim \frac{n\log n}{\Delta_K^2}\frac{1}{M}$ holds, reliable top-$K$ ranking is guaranteed with the sample size $\binom{n}{M}pL \gtrsim \frac{n\log n}{\Delta_K^2}\frac{1}{M}$. This shows that Theorem 3 implies Theorem 2.

Due to the lack of analytical tools for obtaining $\ell_\infty$ errors, we first characterize an upper bound of $\ell_2$ errors with well-known tools. We show that the pointwise error is bounded, in an order-wise manner, by the *same* bound of the $\ell_2$ errors in some restricted regime. See (3) and (4). That is, we obtain the upper bound of $\ell_2$ errors described in Theorem 4 and tightly link it to the bound of $\ell_\infty$ errors we derive in Theorem 3.

**Theorem 4.** *$\ell_2$ norm estimation errors can be upper-bounded by*

$$\frac{\|\hat{\boldsymbol{w}} - \boldsymbol{w}\|_2}{\|\boldsymbol{w}\|_2} \lesssim \sqrt{\frac{n\log n}{\binom{n}{M}pL}}\sqrt{\frac{1}{M}}, \tag{4}$$

*with high probability, where $L \geq \left\lceil c_3\frac{\log n}{\binom{n-1}{M-1}p} \right\rceil$, $p > c_4\frac{\log n}{\binom{n-1}{M-1}}$ and $c_3$ and $c_4$ are some numerical constants.*

Now, let us provide the proof of Theorem 3 assuming for the time being that Theorem 4 is true. We will provide the proof of Theorem 4 in Section 3 once we prove Theorem 3 in Section 2.

## 2 Proof of Theorem 3

**Outline:** The proof of Theorem 3 consists of Lemmas 1, 2 and 3. Lemmas 1 and 2 are straightforward to obtain by applying Hoeffding's inequality stated in Appendix 5.3. Lemma 3, as emphasized, is key; it plays an important role in tightly linking $\ell_2$ and $\ell_\infty$ error bounds. In proving it, we use Janson's inequality (stated in Lemma 4) which describes the concentration behavior of sums of *dependent* random variables, and the $\ell_2$ error bound derived in Theorem 4.

---
**Proof dependencies:**
Theorem 3 ⟵ Lemma 1, Lemma 2, Lemma 3
    Lemma 1 ⟵ Hoeffding's equality (Appendix 5.3)
    Lemma 2 ⟵ Hoeffding's equality
    Lemma 3 ⟵ Janson's equality (Lemma 4), Theorem 4 (Section 3)
---

**Lemma 1.** *Suppose $L \geq 25(1+b)^2\frac{\log n}{\binom{n-1}{M-1}p}$, where $b := \frac{w_{\max}}{w_{\min}}$. Then,*

$$\hat{P}_{ii} < 1 \tag{5}$$

*with probability at least $1 - 2n^{-2}$.*

**Lemma 2.** *For a comparison graph $\mathcal{G} = \left([n], \mathcal{E}^{(M)}\right)$,*

$$\left| \sum_{j:j\neq i} (w_i + w_j)\left(\hat{P}_{ji} - P_{ji}\right) \right| \leq 4w_{\max}\sqrt{\frac{\log n}{\binom{n-1}{M-1}pL}} \tag{6}$$

*with probability at least $1 - 2n^{-2}$.*

**Lemma 3.** *Suppose $p \geq c_1(M-1)\sqrt{\frac{\log n}{\binom{n-1}{M-1}}}$ and $M \geq 3$. Then, in the regime where $n$ is sufficiently large,*

$$\sum_{j:j\neq i} |\hat{w}_j - w_j|\,\hat{P}_{ij} \leq c_5 w_{\max}\sqrt{\frac{\log n}{\binom{n-1}{M-1}pL}} \tag{7}$$

*with probability at least $1 - 2n^{-3c_1^2/50}$, where $c_1$ and $c_5$ are some constants.*

We first assume that these lemmas hold, and proceed to prove Theorem 3. We provide the proofs of these lemmas afterward. Now, let us begin to prove Theorem 3.

**Proof:** To find an upper bound of $\ell_\infty$ errors, we first derive an upper bound, which we will prove very soon, on the pointwise error between the score estimate of item $i$ and the true score, which consists of three terms:

$$|\hat{w}_i - w_i| \leq |\hat{w}_i - w_i|\,\hat{P}_{ii} + \sum_{j:j\neq i} |\hat{w}_j - w_j|\,\hat{P}_{ij} + \left| \sum_{j:j\neq i}(w_i + w_j)\left(\hat{P}_{ji} - P_{ji}\right) \right|. \tag{8}$$

Then, we use the three lemmas stated above, which we prove soon. We consider the regime where $n$ is sufficiently large. For $L \geq \left\lceil 25(1+b)^2\frac{\log n}{\binom{n-1}{M-1}p} \right\rceil$, applying Lemmas 1, 2 and 3 to (8) and solving it, we get

$$|\hat{w}_i - w_i| \leq \frac{1}{1 - \hat{P}_{ii}}(4 + c_5)\sqrt{\frac{\log n}{\binom{n-1}{M-1}pL}} \leq c_6 w_{\max}\sqrt{\frac{n\log n}{\binom{n}{M}pL}}\sqrt{\frac{1}{M}}, \tag{9}$$

where $c_6$ is a constant. This completes the proof of (2).

**Proof of (8):** Since $\hat{\boldsymbol{w}}$ is the stationary distribution of matrix $\hat{P}$, $\hat{\boldsymbol{w}} = \hat{\boldsymbol{P}}\hat{\boldsymbol{w}}$ holds. Thus, for fixed $i$, we get

$$\hat{w}_i = \hat{w}_i\hat{P}_{ii} + \sum_{j:j\neq i}\hat{w}_j\hat{P}_{ij}. \tag{10}$$

Using the fact that random walks on an ideal version of matrix $\hat{\boldsymbol{P}}$ (matrix $\boldsymbol{P}$) are reversible, we get

$$w_i = w_i\left(1 - \sum_{j:j\neq i}P_{ji}\right) + w_i\sum_{j:j\neq i}P_{ji} = w_i\left(1 - \sum_{j:j\neq i}P_{ji}\right) + \sum_{j:j\neq i}w_jP_{ij}$$

$$= \left\{ w_i\hat{P}_{ii} + \sum_{j:j\neq i}w_i\left(\hat{P}_{ji} - P_{ji}\right) \right\} + \left\{ \sum_{j:j\neq i}w_j\hat{P}_{ij} - \sum_{j:j\neq i}w_j\left(\hat{P}_{ij} - P_{ij}\right) \right\}. \tag{11}$$

Using (10) and (11), we get

$$\hat{w}_i - w_i = (\hat{w}_i - w_i)\,\hat{P}_{ii} - \sum_{j:j\neq i}w_i\left(\hat{P}_{ji} - P_{ji}\right) + \sum_{j:j\neq i}(\hat{w}_j - w_j)\,\hat{P}_{ij} + \sum_{j:j\neq i}w_j\left(\hat{P}_{ij} - P_{ij}\right). \tag{12}$$

We note that $\hat{P}_{ji} = \frac{1}{2d_{\max}}\sum_{\mathcal{I}:i,j\in\mathcal{I}}\mathbb{I}\left[\mathcal{I}\in\mathcal{E}^{(M)}\right] - \hat{P}_{ij}$ from $y_{ji} = 1 - y_{ij}$. Similarly, $P_{ji} = \frac{1}{2d_{\max}}\sum_{\mathcal{I}:i,j\in\mathcal{I}}\mathbb{I}\left[\mathcal{I}\in\mathcal{E}^{(M)}\right] - P_{ij}$. Thus, $\hat{P}_{ji} - P_{ji} = -\left(\hat{P}_{ij} - P_{ij}\right)$. Applying this equality and the triangle inequality to (12), we get the relation (8).

## 2.1 Proof of Lemma 1

First, by using Hoeffding's inequality in Appendix 5.3, we get

$$\left| \sum_{j:j\neq i}\left(\hat{P}_{ji} - P_{ji}\right) \right| \leq 2\sqrt{\frac{\log n}{\binom{n-1}{M-1}pL}} \tag{13}$$

with probability at least $1 - 2n^{-2}$. To show this, we represent $\left|\sum_{j:j\neq i}\left(\hat{P}_{ji} - P_{ji}\right)\right|$ as a sum of random variables as follows.

$$\sum_{j:j\neq i}\left(P_{ji} - \hat{P}_{ji}\right) = \sum_{j:j\neq i}\sum_{\mathcal{I}:\{i,j\}\in\phi(\mathcal{I})}\sum_{\ell=1}^{L}\frac{1}{2Ld_{\max}}\left(\frac{w_j}{w_i+w_j} - y_{ji,\mathcal{I}}^{(\ell)}\right) = \frac{1}{2Ld_{\max}}\sum_{j:j\neq i}\sum_{\mathcal{I}:\{i,j\}\in\phi(\mathcal{I})}\sum_{\ell=1}^{L}\left(-y_{ji,\mathcal{I}}^{(\ell)} + \frac{w_j}{w_i+w_j}\right). \tag{14}$$

Let $X := \sum_{\mathcal{I}:i\in\mathcal{I}}\sum_{\ell=1}^{L}\sum_{m:\{i,m\}\in\phi(\mathcal{I})}\left(y_{mi,\mathcal{I}}^{(\ell)} - \frac{w_m}{w_m+w_i}\right)$. Applying Hoeffding's inequality to $X$, we get

$$\Pr\left[|X| \geq t\Big|\mathcal{E}^{(M)}\right] \overset{(a)}{\leq} 2\exp\left(-\frac{2t^2}{\sum_{\mathcal{I}:i\in\mathcal{I}}\sum_{\ell=1}^{L}2^2}\right) \leq 2\exp\left(-\frac{2t^2}{4d_iL}\right), \tag{15}$$

where $(a)$ follows by the fact that $\sum_{m:\{i,m\}\in\phi(\mathcal{I})}y_{mi,\mathcal{I}}^{(\ell)}$ varies from 0 to 2; the rest follows by straightforward computation. Finally, choosing $t = 2\sqrt{2Ld_i\log n}$, we can show that $\left|\sum_{j:j\neq i}\left(\hat{P}_{ji} - P_{ji}\right)\right| \leq \sqrt{\frac{2\log n}{Ld_i}}$ holds with probability at least $1-2n^{-4}$. This leads to $\left|\sum_{j:j\neq i}\left(\hat{P}_{ji} - P_{ji}\right)\right| \leq 2\sqrt{\frac{\log n}{\binom{n-1}{M-1}pL}}$ since $\frac{1}{2}\binom{n-1}{M-1}p \leq d_i \leq \frac{3}{2}\binom{n-1}{M-1}p$ holds with probability at least $1 - 2n^{-2}$ by Lemma 7 for sufficiently large $p$, which is the regime in which we are interested.

Using (13), we get

$$\hat{P}_{ii} = 1 - \sum_{j:j\neq i}\hat{P}_{ji} \leq 1 - \sum_{j:j\neq i}P_{ji} + 2\sqrt{\frac{\log n}{\binom{n-1}{M-1}pL}}. \tag{16}$$

We let $b = \frac{w_{\max}}{w_{\min}}$. From the definition of $P_{ji}$,

$$\sum_{j:j\neq i}P_{ji} = \sum_{j:j\neq i}\frac{1}{2d_{\max}}\sum_{\mathcal{I}:\{i,j\}\in\phi(\mathcal{I})}\frac{1}{1+\frac{w_j}{w_i}} \geq \frac{1}{2d_{\max}}\sum_{j:j\neq i}\sum_{\mathcal{I}:\{i,j\}\in\phi(\mathcal{I})}\frac{1}{1+b} \tag{17}$$

$$= \frac{1}{2d_{\max}}\sum_{\mathcal{I}:i\in\mathcal{I}}\sum_{j:\{i,j\}\in\phi(\mathcal{I})}\frac{1}{1+b} = \frac{1}{2d_{\max}}\sum_{\mathcal{I}:i\in\mathcal{I}}\frac{2}{1+b} \geq \frac{d_{\min}}{d_{\max}}\frac{1}{1+b} \geq \frac{1}{3(1+b)}. \tag{18}$$

Putting (17) into (16), we get

$$\hat{P}_{ii} \leq 1 - \frac{1}{3(1+b)} + 2\sqrt{\frac{\log n}{\binom{n-1}{M-1}pL}}. \tag{19}$$

Choosing $L \geq 25(1+b)^2\frac{\log n}{\binom{n-1}{M-1}p}$, we complete the proof of Lemma 1.

## 2.2 Proof of Lemma 2

By using a slightly modified Hoeffding's inequality used to show (13), we get

$$\left|\sum_{j:j\neq i}(w_i+w_j)\left(\hat{P}_{ji} - P_{ji}\right)\right| = \left|\frac{1}{2Ld_{\max}}\sum_{j:j\neq i}\sum_{\mathcal{I}:\{i,j\}\in\phi(\mathcal{I})}\sum_{\ell=1}^{L}\left[(w_i+w_j)\left(-y_{ji,\mathcal{I}}^{(\ell)} + \frac{w_j}{w_i+w_j}\right)\right]\right| \leq 4w_{\max}\sqrt{\frac{\log n}{\binom{n-1}{M-1}pL}} \tag{20}$$

with probability at least $1 - 2n^{-2}$. We can see that each random variable $\left(-y_{ji,\mathcal{I}}^{(\ell)} + \frac{w_j}{w_i+w_j}\right)$ in (14) is replaced by $(w_i+w_j)\left(-y_{ji,\mathcal{I}}^{(\ell)} + \frac{w_j}{w_i+w_j}\right)$. Thus, the range of each random variable is extended by at most $2w_{\max}$. Applying a similar line of steps in (15), we get (20).

## 2.3 Proof of Lemma 3

**Outline:** As mentioned at the beginning, Lemma 3 plays a key role in linking the $\ell_2$ error bound in Theorem 4 to the $\ell_\infty$ error bound in Theorem 3, leading them to be on the same order. In doing so, we have sums of *dependent* random variables to handle, thus we make use of Janson's inequality (Janson 2002) stated in Lemma 4.

**Proof:** First, let us define $B$ as follows.

$$B := \sum_{j:j\neq i} |\hat{w}_j - w_j| \, \hat{P}_{ij}. \tag{21}$$

By Lemma 2, with probability at least $1 - 2n^{-2}$,

$$\left| \sum_{j:j\neq i} (w_i + w_j)\left( \hat{P}_{ji} - P_{ji} \right) \right| \leq 4w_{\max}\sqrt{\frac{\log n}{\binom{n-1}{M-1}pL}}. \tag{22}$$

Putting (8) with (22) into (21), we get

$$B \leq \sum_{j:j\neq i} |\hat{w}_j - w_j| \, \hat{P}_{jj}\hat{P}_{ij} + 4w_{\max}\sqrt{\frac{\log n}{\binom{n-1}{M-1}pL}} \sum_{j:j\neq i} \hat{P}_{ij} + \sum_{j:j\neq i}\sum_{k:k\neq j} |\hat{w}_k - w_k| \, \hat{P}_{jk}\hat{P}_{ij}. \tag{23}$$

We simplify the last two terms. The first of the two is straightforward. The definition of $\hat{P}_{ij}$ gives $\sum_{j:j\neq i} \hat{P}_{ij} \leq 1$. The last term needs some extra efforts. For the time being, we state the following, whose proof which makes use of Janson's inequality stated in Lemma 4 will soon be provided.

$$\sum_{j:j\neq i}\sum_{k:k\neq j} |\hat{w}_k - w_k| \, \hat{P}_{jk}\hat{P}_{ij} \leq c_7 \, \|\hat{\boldsymbol{w}} - \boldsymbol{w}\|_2 \sqrt{\frac{1}{n}}. \tag{24}$$

Putting $\sum_{j:j\neq i} \hat{P}_{ij} \leq 1$ and (24) into (23), we get

$$B \leq \sum_{j:j\neq i} |\hat{w}_j - w_j| \, \hat{P}_{jj}\hat{P}_{ij} + 4w_{\max}\sqrt{\frac{\log n}{\binom{n-1}{M-1}pL}} + c_7\sqrt{\frac{1}{n}}\,\|\hat{\boldsymbol{w}} - \boldsymbol{w}\|_2. \tag{25}$$

By Lemma 1, we can find a constant $\beta$ such that $\hat{P}_{jj} \leq \beta < 1$ for all $j$. Using such $\beta$, we get

$$B \leq \beta B + 4w_{\max}\sqrt{\frac{\log n}{\binom{n-1}{M-1}pL}} + c_7\sqrt{\frac{1}{n}}\,\|\hat{\boldsymbol{w}} - \boldsymbol{w}\|_2. \tag{26}$$

Here, we use an upper bound on $\frac{\|\hat{\boldsymbol{w}} - \boldsymbol{w}\|_2}{\|\boldsymbol{w}\|_2}$ derived by Theorem 4. Theorem 4 states that when $L \geq c_3 \frac{\log n}{\binom{n-1}{M-1}p}$, for a constants $c_8$,

$$\frac{\|\hat{\boldsymbol{w}} - \boldsymbol{w}\|_2}{\|\boldsymbol{w}\|_2} \leq c_8\sqrt{\frac{\log n}{\binom{n-1}{M-1}pL}}. \tag{27}$$

Using $\|\boldsymbol{w}\|_2 \leq \sqrt{n}\,\|\boldsymbol{w}\|_\infty = \sqrt{n}w_{\max}$, we get

$$\|\hat{\boldsymbol{w}} - \boldsymbol{w}\|_2 \leq \sqrt{n}w_{\max}c_8\sqrt{\frac{\log n}{\binom{n-1}{M-1}pL}}. \tag{28}$$

Putting (28) into (26) and solving it, we get

$$B \leq \frac{1}{1-\beta} w_{\max} (c_7 c_8 + 4) \sqrt{\frac{\log n}{\binom{n-1}{M-1} pL}} = c_5 w_{\max} \sqrt{\frac{\log n}{\binom{n-1}{M-1} pL}}. \tag{29}$$

From the definition of $B$, we complete the proof of Lemma 3.

**Proof of (24)**: By changing the order of the summations and the Cauchy-Schwarz inequality, we get

$$\sum_{j:j\neq i} \sum_{k:k\neq j} |\hat{w}_k - w_k| \hat{P}_{jk} \hat{P}_{ij} = \sum_k |\hat{w}_k - w_k| \sum_{j:j\notin\{i,k\}} \hat{P}_{jk} \hat{P}_{ij} \leq \|\hat{\boldsymbol{w}} - \boldsymbol{w}\|_2 \sqrt{\sum_k \left( \sum_{j:j\notin\{i,k\}} \hat{P}_{jk} \hat{P}_{ij} \right)^2}. \tag{30}$$

When we show that $\sum_{j:j\notin\{i,k\}} \hat{P}_{jk} \hat{P}_{ij} \leq \frac{c_7}{n}$ holds, we can finally conclude that $\sum_{j:j\neq i} \sum_{k:k\neq j} |\hat{w}_k - w_k| \hat{P}_{jk} \hat{P}_{ij} \leq c_7 \sqrt{\frac{1}{n}}$.

Now, let us prove that $\sum_{j:j\notin\{i,k\}} \hat{P}_{jk} \hat{P}_{ij} \leq \frac{c_7}{n}$ holds. From the definitions of $\hat{P}_{jk}$ and $\hat{P}_{ij}$, we can bound the term $\sum_{j:j\neq i,k} \hat{P}_{jk} \hat{P}_{ij}$ as follows. First, we can expand $\sum_{j:j\neq i,k} \hat{P}_{ij} \hat{P}_{jk}$ and bound it as follows.

$$\sum_{j:j\neq i,k} \hat{P}_{ij} \hat{P}_{jk} = \frac{1}{4d_{\max}^2} \sum_{j:j\neq i,k} \left[ \left( \sum_{\mathcal{I}_1:\{i,j\}\in\phi(\mathcal{I}_1)} y_{ij,\mathcal{I}_1} \right) \left( \sum_{\mathcal{I}_2:\{k,j\}\in\phi(\mathcal{I}_2)} y_{jk,\mathcal{I}_2} \right) \right] \tag{31}$$

$$\leq \frac{1}{4d_{\max}^2} \sum_{j:j\neq i,k} \left[ \left( \sum_{\mathcal{I}_1:i,j\in\mathcal{I}_1} \mathbb{I}\left[\{i,j\}\in\phi(\mathcal{I}_1)\right] \right) \left( \sum_{\mathcal{I}_2:j,k\in\mathcal{I}_2} \mathbb{I}\left[\{j,k\}\in\phi(\mathcal{I}_2)\right] \right) \right] \tag{32}$$

$$= \frac{1}{4d_{\max}^2} \sum_{j:j\neq i,k} \sum_{\mathcal{I}_1:i,j\in\mathcal{I}_1} \sum_{\mathcal{I}_2:j,k\in\mathcal{I}_2} \mathbb{I}\left[\{i,j\}\in\phi(\mathcal{I}_1)\right] \mathbb{I}\left[\{j,k\}\in\phi(\mathcal{I}_2)\right] = \frac{1}{4d_{\max}^2} \sum_{j:j\neq i,k} \sum_{\mathcal{I}_1:i,j\in\mathcal{I}_1} \sum_{\mathcal{I}_2:j,k\in\mathcal{I}_2} X_{\mathcal{I}_1\mathcal{I}_2}, \tag{33}$$

where $X_{\mathcal{I}_1\mathcal{I}_2} \sim \text{Bern}\left(\frac{p^2}{(M-1)^2}\right)$ when $\mathcal{I}_1 \neq \mathcal{I}_2$ and $X_{\mathcal{I}_1\mathcal{I}_2} \sim \text{Bern}\left(\frac{p}{M-1}\right)$ when $\mathcal{I}_1 = \mathcal{I}_2$. It follows by the fact that hyper-edge $\mathcal{I}_1$ is chosen with probability $p$ and a partial pairwise comparison between $i$ and $j$ is chosen with probability $\frac{1}{M-1}$. See (101) for details.

Note that $X_{\mathcal{I}_1\mathcal{I}_2}$ and $X_{\mathcal{I}_1\mathcal{I}_3}$, concerning the same hyper-edge $\mathcal{I}_1$, are dependent random variables. Computing the expectation of this sum of dependent random variables, we get

$$\mathbb{E}\left[ \sum_{j:j\neq i,k} \hat{P}_{ij} \hat{P}_{jk} \right] = \frac{1}{4d_{\max}^2} \sum_{j:j\neq i,k} \sum_{\mathcal{I}_1:i,j\in\mathcal{I}_1} \sum_{\mathcal{I}_2:j,k\in\mathcal{I}_2} \mathbb{E}\left[X_{\mathcal{I}_1\mathcal{I}_2}\right] \tag{34}$$

$$= \frac{1}{4d_{\max}^2} \left( (\text{\# of r.v's}: \mathcal{I}_1 \neq \mathcal{I}_2) \left(\frac{p}{M-1}\right)^2 + (\text{\# of r.v's}: \mathcal{I}_1 = \mathcal{I}_2) \frac{p}{M-1} \right) \tag{35}$$

$$\overset{(a)}{\leq} \frac{1}{4d_{\max}^2} \left( (n-1)\binom{n-2}{M-2}^2 \left(\frac{p}{M-1}\right)^2 + (n-1)\binom{n-2}{M-2}\frac{p}{M-1} \right) \overset{(b)}{\leq} \frac{2}{4d_{\max}^2}(n-1)\frac{1}{(n-1)^2}\left(\frac{n-1}{M-1}\right)^2 p^2 \overset{(c)}{\leq} \frac{3}{n}, \tag{36}$$

where $(a)$ follows by the facts that one can bound the number of cases where $\mathcal{I}_1 \neq \mathcal{I}_2$ by $(n-1)\binom{n-2}{M-2}^2$, and that one can bound the number of cases where $\mathcal{I}_1 = \mathcal{I}_2$ by $(n-1)\binom{n-2}{M-2}$; $(b)$[1] follows by the fact that $(n-1)\binom{n-2}{M-2}^2 \frac{p^2}{(M-1)^2} \geq$

_______________

[1] The provided steps are tailored for $M > 2$ where our algorithm that features sample breaking can be employed, but do not hold for $M = 2$ where sample breaking does not come into the picture. However, following a similar line of steps with some simple modifications, one can show that an upper bound on the expectation is also on the order of $n^{-1}$, as in (Chen and Suh 2015).

$(n-1)\binom{n-2}{M-2}\frac{p}{M-1}$ for $M > 2$ and $p \geq c_3(M-1)\sqrt{\frac{\log n}{\binom{n-1}{M-1}}}$; (c) follows by the fact that $d_{\max} \geq \frac{1}{2}\binom{n-1}{M-1}p$, which can be shown by Lemma 7 that describes the concentration behavior of sums of independent random variables. This bound tells us that once $\sum_{j:j\neq i,k}\hat{P}_{ij}\hat{P}_{jk}$ concentrates to its expectation, we can prove (24).

To show that $\sum_{j:j\neq i,k}\hat{P}_{ij}\hat{P}_{jk}$ concentrates to its expectation, we apply the concentration inequality for a sum of *dependent* random variables, called Janson's inequality (Janson 2002). Here we provide the statement of Janson's inequality.

**Lemma 4** (Janson's inequality (Janson 2002)). *Suppose that* $\tilde{X} = \sum_{i=1}^N \tilde{X}_i$ *with* $\left|\tilde{X}_i - \mathbb{E}\left[\tilde{X}_i\right]\right| \leq C$ *for some* $C > 0$ *and all* $i$. *Then, for* $t \geq 0$,

$$\Pr\left[\left|\tilde{X} - \mathbb{E}\left[\tilde{X}\right]\right| \geq t\right] \leq 2\exp\left(-\frac{8t^2}{25d\left(\sum_{i=1}^N Var\left[\tilde{X}_i\right] + Ct/3\right)}\right), \tag{37}$$

*where* $d$ *is the maximum number of random variables dependent of* $\tilde{X}_i$ *over* $i$.

To get an upper bound of $\frac{1}{4d_{\max}^2}\sum_{j:j\neq i,k}\sum_{\mathcal{I}_1:i,j\in\mathcal{I}_1}\sum_{\mathcal{I}_2:j,k\in\mathcal{I}_2}X_{\mathcal{I}_1\mathcal{I}_2}$ in (33) by applying Janson's inequality, let us define $\tilde{X}$ as follows.

$$\tilde{X} := \sum_{j:j\neq i,k}\sum_{\mathcal{I}_1:i,j\in\mathcal{I}_1}\sum_{\mathcal{I}_2:j,k\in\mathcal{I}_2}X_{\mathcal{I}_1\mathcal{I}_2}. \tag{38}$$

Once we show that $\left|\tilde{X} - \mathbb{E}\left[\tilde{X}\right]\right| \leq (n-2)\binom{n-2}{M-2}^2 p^2$ holds with high probability by using Janson's inequality, we can conclude that $\sum_{j:j\neq i,k}\hat{P}_{ij}\hat{P}_{jk} \leq \frac{c_7}{n}$ holds with high probability.

$$\Pr\left[\left|\tilde{X} - \mathbb{E}\left[\tilde{X}\right]\right| \geq t\right] \leq 2\exp\left(-\frac{8t^2}{25d\left(\sum_{j:j\neq i,k}\sum_{\mathcal{I}_1:i,j\in\mathcal{I}_1}\sum_{\mathcal{I}_2:j,k\in\mathcal{I}_2}Var\left[X_{\mathcal{I}_1\mathcal{I}_2}\right] + Ct/3\right)}\right) \tag{39}$$

$$\overset{(a)}{\leq} 2\exp\left(-\frac{8t^2}{25d\left(\sum_{j:j\neq i,k}\sum_{\mathcal{I}_1:i,j\in\mathcal{I}_1}\sum_{\mathcal{I}_2:j,k\in\mathcal{I}_2}\mathbb{E}\left[X_{\mathcal{I}_1\mathcal{I}_2}\right] + Ct/3\right)}\right) \tag{40}$$

$$\overset{(b)}{\leq} 2\exp\left(-\frac{8t^2}{50(M-1)\binom{n-2}{M-2}\left(\sum_{j:j\neq i,k}\sum_{\mathcal{I}_1:i,j\in\mathcal{I}_1}\sum_{\mathcal{I}_2:j,k\in\mathcal{I}_2}\mathbb{E}\left[X_{\mathcal{I}_1\mathcal{I}_2}\right] + t/3\right)}\right) \tag{41}$$

$$\overset{(c)}{\leq} 2\exp\left(-\frac{8t^2}{50(M-1)\binom{n-2}{M-2}\left(\frac{1}{n}\binom{n-1}{M-1}^2 p^2 + t/3\right)}\right) \overset{(d)}{\leq} 2\exp\left(-\frac{8\left(\frac{1}{n}\binom{n-1}{M-1}^2 p^2\right)^2}{100(M-1)\binom{n-2}{M-2}\left(\frac{1}{n}\binom{n-1}{M-1}^2 p^2 + \frac{1}{3n}\binom{n-1}{M-1}^2 p^2\right)}\right) \tag{42}$$

$$\overset{(e)}{\leq} 2\exp\left(-\frac{3\binom{n-1}{M-1}p^2}{50(M-1)^2}\right) \leq 2n^{-\frac{3c_1^2}{50}} \tag{43}$$

where (a) follows by the fact that $Var\left[X_{\mathcal{I}_1\mathcal{I}_2}\right] = \left(\frac{p}{M-1}\right)^2\left(1 - \left(\frac{p}{M-1}\right)^2\right) \leq \left(\frac{p}{M-1}\right)^2 = \mathbb{E}\left[X_{\mathcal{I}_1\mathcal{I}_2}\right]$; (b) follows by the fact that $d \leq 2(M-1)\binom{n-2}{M-2}$. Let us further elaborate this step. Suppose we have $i$ and $k$ given. Let us fix $\mathcal{I}_1$ and choose $j \neq i$. Then for chosen $j$, there are $\binom{n-2}{M-2}$ distinct $\mathcal{I}_2$'s since we can choose $M-2$ items and combine them with $j$ and given $k$ to form $\mathcal{I}_2$. Also, there are $M-1$ ways to pick $j \neq i$ to form the previously fixed $\mathcal{I}_1$ since $j$ can be the items in $\mathcal{I}_1$ except given $i$. These two facts amount to $(M-1)\binom{n-2}{M-2}$. Changing the roles of $\mathcal{I}_1$ and $\mathcal{I}_2$, we get $d \leq 2(M-1)\binom{n-2}{M-2}$; (c) follows by the fact we can bound $\mathbb{E}\left[\tilde{X}\right]$ as in (36); (d) follows by choosing $t = (n-2)\binom{n-2}{M-2}^2 p^2$; (e) follows by the fact that $p > c_1(M-1)\sqrt{\frac{\log n}{\binom{n-1}{M-1}}}$.

# 3  Proof of Theorem 4

**Outline:** The line of steps we follow to prove Theorem 4 is similar to that taken in (Negahban, Oh and Shah 2016). To be more specific, the base inequality from which we build on to derive an upper bound of $\ell_2$ errors (48) is derived in the proof of Lemma 2 in (Negahban, Oh and Shah 2016). To prove Theorem 4, we introduce two lemmas: Lemmas 5 and 6. Lemma 5 corresponds to Lemmas 3 and 5 in (Negahban, Oh and Shah 2016), and Lemma 6 corresponds to Lemma 4 therein. The difference largely comes from the fact that required calculations to derive our lemmas need to be more involved, as we consider a more general model. Aside from this difference, the proof of Theorem 4 mostly adopts an existing technique that derives $\ell_2$ error bounds.

---
**Proof dependencies:**
Theorem 4 $\longleftarrow$ Lemma 5, Lemma 6
  Lemma 5 $\longleftarrow$ Equation (56), Equation (62)
    Equation (56) $\longleftarrow$ Hoeffding's inequality (Appendix 5.3)
    Equation (62) $\longleftarrow$ Matrix Bernstein inequality (Appendix 5.5)
  Lemma 6 $\longleftarrow$ Equation (79), Equation (80)
    Equation (79)
    Equation (80) $\longleftarrow$ Equation (88), Equation (89), Equation (90)
      Equation (88) $\longleftarrow$ Hölder's inequality (Appendix 5.4)
      Equation (89) $\longleftarrow$ Hölder's inequality
      Equation (90) $\longleftarrow$ Matrix Bernstein inequality

---

**Lemma 5.** *Suppose that $p \geq c_4 \frac{\log n}{\binom{n-1}{M-1}}$, where $c_4$ is sufficiently large. Then,*

$$\|\Delta\|_2 \leq 10\sqrt{\frac{\log n}{\binom{n-1}{M-1}pL}} \tag{44}$$

*with probability at least $1 - 2n^{-3/5}$.*

**Lemma 6.** *Suppose that $L \geq c_3 \frac{\log n}{\binom{n-1}{M-1}p}$. Then,*

$$h(P) \geq \frac{1}{270b^2} \tag{45}$$

*with probability at least $1 - 2n^{-1/15}$, where $c_3$ is some numerical constant.*

We first assume that these lemmas hold, and proceed to prove Theorem 4. We provide the proofs of these lemmas afterward. Now, let us begin to prove Theorem 4.

**Proof:** From the definition of $P$ in Section 2 and the algorithm description in Section 3.1 in the main paper, we get

$$\boldsymbol{w} = P\boldsymbol{w}, \quad \hat{\boldsymbol{w}} = \hat{P}\hat{\boldsymbol{w}}. \tag{46}$$

Using two balance equations in (46), we get

$$\hat{\boldsymbol{w}} - \boldsymbol{w} = \hat{P}\hat{\boldsymbol{w}} - P\boldsymbol{w} = \hat{P}\left(\hat{\boldsymbol{w}} - \boldsymbol{w}\right) + \left(\hat{P} - P\right)\boldsymbol{w}. \tag{47}$$

From (47), we can get the $\ell_2$ error of estimate $\hat{\boldsymbol{w}}$ as follows.

$$\|\hat{\boldsymbol{w}} - \boldsymbol{w}\|_2 \leq \left(1 - h(P) + \sqrt{b}\,\|\Delta\|_2\right)\sqrt{b}\,\|\hat{\boldsymbol{w}} - \boldsymbol{w}\|_2 + \sqrt{b}\,\|\Delta\|_2\,\|\boldsymbol{w}\|_2, \tag{48}$$

where $h(P)$ is the spectral gap of matrix $P$, and the equality follows by letting $\Delta := \hat{P} - P$. The proof of (48) is derived in the proof of Lemma 2 in (Negahban, Oh and Shah 2016).

We can see that, for (48) to get a proper upper bound of $\|\hat{\boldsymbol{w}} - \boldsymbol{w}\|_2$, the term $1 - h(P) + \sqrt{b}\,\|\Delta\|_2$ needs to be less than one. To safely guarantee it, we can impose the following condition:

$$\sqrt{b}\,\|\Delta\|_2 \leq \frac{h(P)}{2}. \tag{49}$$

We can obtain an upper bound on $\|\Delta\|_2$ that holds with high probability and a lower bound on $h(P)$. The first corresponds to Lemma 5 and the second corresponds to Lemma 6. We will soon provide their proofs.

From (49) and (44), we get

$$10\sqrt{b}\sqrt{\frac{\log n}{\binom{n-1}{M-1}pL}} \leq \frac{h(P)}{2} \iff L \geq \frac{400b}{h(P)^2}\frac{\log n}{\binom{n-1}{M-1}p}, \tag{50}$$

and from (45) and (50), we get

$$L \geq \left\lceil c_3 \frac{\log n}{\binom{n-1}{M-1}p} \right\rceil, \tag{51}$$

where $c_3 := 29160000b^5$.

Solving the equation (48) and replacing $\sqrt{b}\,\|\Delta\|_2$ and $\|\Delta\|_2$ by (49) and (44) respectively, we get

$$\frac{\|\hat{\boldsymbol{w}} - \boldsymbol{w}\|_2}{\|\boldsymbol{w}\|_2} \leq \frac{1}{h(P)/2}\sqrt{b}\left(10\sqrt{\frac{\log n}{\binom{n-1}{M-1}pL}}\right). \tag{52}$$

Replacing $h(P)$ with the lower bound in (45) and by direct computation, we get

$$\frac{\|\hat{\boldsymbol{w}} - \boldsymbol{w}\|_2}{\|\boldsymbol{w}\|_2} \leq 1350b^{3/2}\sqrt{\frac{\log n}{\binom{n-1}{M-1}pL}} = 1350b^{3/2}\sqrt{\frac{n\log n}{M\binom{n}{M}pL}} \lesssim \sqrt{\frac{n\log n}{\binom{n}{M}pL}}\sqrt{\frac{1}{M}}, \tag{53}$$

where $p > \frac{c_4 \log n}{\binom{n-1}{M-1}}$ and $L \geq \left\lceil c_3 \frac{\log n}{\binom{n-1}{M-1}p} \right\rceil$. This provides an upper bound on $\ell_2$ errors.

## 3.1 Proof of Lemma 5

**Outline:** Applying the triangle inequality, we get $\|\Delta\|_2 \leq \|\Delta_D\|_2 + \|\Delta_O\|_2$, where $\Delta := \Delta_D + \Delta_O$ and $\Delta_D$ is the matrix whose diagonal entries are equal to those of $\Delta$ while the other entries are zero. (Hence, we refer to $\Delta_D$ as the diagonal matrix of $\Delta$, and $\Delta_O$ as the off-diagonal matrix of $\Delta$.) To show that (44) holds, we will bound $\|\Delta_D\|_2$ and $\|\Delta_O\|_2$ separately.

Firstly, bounding the diagonal matrix $\|\Delta_D\|_2$ in (56) is straightforward by Hoeffding's inequality.

On the other hand, bounding the off-diagonal matrix $\|\Delta_O\|_2$ in (62) needs some extra efforts. We primarily apply the matrix Bernstein inequality stated in Appendix 5.5 to bound $\|\Delta_O\|_2$. To apply it, we need to obtain an equality and an inequality, (59) and (60) respectively, which are needed as parameters in the matrix Bernstein inequality. To prove (59), we use the Courant-Fischer theorem stated in Appendix 5.6. To prove (60), we use the matrix version of Hölder's inequality stated in Appendix 5.4 in addition to the Courant-Fischer theorem.

---
**Proof dependencies:**

Lemma 5 ⟵ Equation (56), Equation (62)

   Equation (56) ⟵ Hoeffding's inequality (Appendix 5.3)

   Equation (62) ⟵ Matrix Bernstein inequality (Appendix 5.5)

     Matrix Bernstein inequality ⟵ Parameters (59) and (60)

       Parameter (59) ⟵ Courant-Fischer theorem (Appendix 5.6)

       Parameter (60) ⟵ Courant-Fischer theorem, Hölder inequality (Appendix 5.4)

---

### 3.1.1 Bound on $\|\Delta_D\|_2$:

By the definition of $\Delta_D$, we begin with the following equality:

$$\|\Delta_D\|_2 = \max_i \left| \hat{P}_{ii} - P_{ii} \right|. \tag{54}$$

Modifying $\hat{P}_{ii} - P_{ii}$, we get $\hat{P}_{ii} - P_{ii} = \sum_{j:j\neq i} \left( P_{ji} - \hat{P}_{ji} \right)$. Applying Hoeffding's inequality used in (13), we get

$$\left| \hat{P}_{ii} - P_{ii} \right| = \left| \sum_{j:j\neq i} \left( P_{ji} - \hat{P}_{ji} \right) \right| \leq 2\sqrt{\frac{\log n}{\binom{n-1}{M-1}pL}} \tag{55}$$

with probability at least $1 - 2n^{-2}$.

Therefore,

$$\|\Delta_D\|_2 \leq 2\sqrt{\frac{\log n}{\binom{n-1}{M-1}pL}}. \tag{56}$$

### 3.1.2 Bound on $\|\Delta_O\|_2$

To obtain a bound on $\|\Delta_O\|_2$, we use the matrix Bernstein inequality (Tropp 2011) in Appendix 5.5.

To apply the matrix Bernstein inequality above to $\|\Delta_O\|_2$, we first need to decompose $\|\Delta_O\|_2$ into the sum of independent, random and self-adjoint matrices. To meet the independence condition, we define $\Delta_{\mathcal{I}}^{(\ell)}$ as follows.

$$\Delta_{\mathcal{I}}^{(\ell)} := \sum_{\{i,j\}\in\phi(\mathcal{I})} \left( e_i e_j^T - e_j e_i^T \right) \left( \frac{1}{2d_{\max}} y_{ij,\mathcal{I}}^{(\ell)} - \frac{1}{2d_{\max}} \frac{w_i}{w_i + w_j} \right) \quad \text{for } \mathcal{I} \in \mathcal{E}^{(M)}. \tag{57}$$

Then, $\Delta_O = \sum_{\mathcal{I}\in\mathcal{E}^{(M)}} \sum_{\ell=1}^{L} \Delta_{\mathcal{I}}^{(\ell)}$ holds, where all $\Delta_{\mathcal{I}}^{(\ell)}$'s are mutually independent. Furthermore, to meet the self-adjoint condition, we define $\tilde{\Delta}_{\mathcal{I}}^{(\ell)}$ as follows.

$$\tilde{\Delta}_{\mathcal{I}}^{(\ell)} := \begin{bmatrix} 0 & \Delta_{\mathcal{I}}^{(\ell)} \\ \left( \Delta_{\mathcal{I}}^{(\ell)} \right)^T & 0 \end{bmatrix}. \tag{58}$$

Note that $\|\Delta_O\|_2 = \left\| \sum_{\mathcal{I}\in\mathcal{E}^{(M)}} \sum_{\ell=1}^{L} \Delta_{\mathcal{I}}^{(\ell)} \right\|_2 = \left\| \sum_{\mathcal{I}\in\mathcal{E}^{(M)}} \sum_{\ell=1}^{L} \tilde{\Delta}_{\mathcal{I}}^{(\ell)} \right\|_2$. Now, to get an upper bound on $\|\Delta_O\|_2$, we need to compute the two parameters $R$ and $\sigma^2$ that appear in (188) in Appendix 5.5. For now, let us assume that the following holds, of which we will provide proofs soon.

$$R = \frac{1}{Ld_{\max}}, \tag{59}$$

$$\sigma^2 \leq \frac{6}{Ld_{\max}}. \tag{60}$$

Then, we get

$$\Pr\left[ \|\Delta_O\|_2 \geq t \right] \leq 2n \exp\left( \frac{-t^2/2}{(6/Ld_{\max}) + (t/Ld_{\max})} \right) \overset{(a)}{=} 2n \exp\left( \frac{-16\log n}{6 + 4\sqrt{\frac{\log n}{Ld_{\max}}}} \right) \overset{(b)}{\leq} 2n \exp\left( \frac{-16\log n}{6 + 4\sqrt{\frac{2}{3}}} \right) \leq 2n^{-3/5}, \tag{61}$$

where $(a)$ follows by choosing $t = 4\sqrt{\frac{\log n}{L d_{\max}}}$; $(b)$ follows by the fact that $L \geq 1$ and $d_{\max} \geq \frac{1}{2}\binom{n-1}{M-1}p \geq \frac{3}{2}\log n$ holds by Lemma 7. Therefore, we get $\|\Delta_O\|_2 \leq 4\sqrt{\frac{\log n}{L d_{\max}}}$ with probability at least $1 - 2n^{-3/5}$.

Using the fact that $d_{\max} \geq \frac{1}{2}\binom{n-1}{M-1}p$ by Lemma 7,

$$\|\Delta_O\|_2 \leq 4\sqrt{\frac{2\log n}{\binom{n-1}{M-1}pL}} \tag{62}$$

with probability at least $1 - 2n^{-3/5}$. This bound on $\|\Delta_O\|_2$, together with the bound on $\|\Delta_D\|_2$ shown earlier, we can get the desired bound on $\|\Delta\|_2$.

As previously mentioned, we now provide the proofs of (59) and (60).

**Proof of (59):** Using the Courant-Fischer theorem in Appendix 5.6,

$$\left\|\tilde{\Delta}_{\mathcal{I}}^{(\ell)}\right\|_2 = \max_{\|v\|_2 = 1} \left| v^T \tilde{\Delta}_{\mathcal{I}}^{(\ell)} v \right|, \tag{63}$$

where $v \in \mathbb{R}^{2n}$. Let us assume that $v^T = \left[x^T, y^T\right]$ where $x, y \in \mathbb{R}^n$. To get an upper bound on $\left\|\tilde{\Delta}_{\mathcal{I}}^{(\ell)}\right\|_2$, we will first derive an upper bound on $\left| v^T \tilde{\Delta}_{\mathcal{I}}^{(\ell)} v \right|$ as follows:

$$\left| v^T \tilde{\Delta}_{\mathcal{I}}^{(\ell)} v \right| = \left| \begin{bmatrix} x^T y^T \end{bmatrix} \begin{bmatrix} 0 & \Delta_{\mathcal{I}}^{(\ell)} \\ \left(\Delta_{\mathcal{I}}^{(\ell)}\right)^T & 0 \end{bmatrix} \begin{bmatrix} x \\ y \end{bmatrix} \right| = \left| y^T \left(\Delta_{\mathcal{I}}^{(\ell)}\right)^T x + x^T \Delta_{\mathcal{I}}^{(\ell)} y \right| \overset{(a)}{=} 2 \sum_{i=1}^{n}\sum_{j=1}^{n} \left| \left(\Delta_{\mathcal{I}}^{(\ell)}\right)_{ij} \right| x_i y_j \tag{64}$$

$$\overset{(b)}{\leq} \sum_{i=1}^{n}\sum_{j=1}^{n} \left| \left(\Delta_{\mathcal{I}}^{(\ell)}\right)_{ij} \right| (x_i^2 + y_j^2) \overset{(c)}{\leq} \sum_{i=1}^{n}\sum_{j=1}^{n} \frac{1}{2d_{\max}L} \mathbb{I}\left[\{i,j\} \in \phi(\mathcal{I})\right] (x_i^2 + y_j^2) \overset{(d)}{=} \frac{1}{2d_{\max}L} 2 \left( \sum_{i:i\in\mathcal{I}} x_i^2 + \sum_{j:j\in\mathcal{I}} y_j^2 \right) \leq \frac{1}{d_{\max}L}\|v\|_2, \tag{65}$$

where $(a)$ follows by the fact that $y^T\left(\Delta_{\mathcal{I}}^{(\ell)}\right)^T x = \left(x^T\Delta_{\mathcal{I}}^{(\ell)}y\right)^T$; $(b)$ follows by the inequality of arithmetic and geometric means; $(c)$ follows by the definition of $\Delta_{\mathcal{I}}^{(\ell)}$ and the fact that $\left| y_{ij,\mathcal{I}} - \frac{w_i}{w_i+w_j} \right| \leq 1$; $(d)$ follows by the fact that in a formed circular permutation, an item is adjacent to two items. Therefore, by the Courant-Fischer theorem, we can get the desired bound $\left\|\tilde{\Delta}_{\mathcal{I}}^{(\ell)}\right\|_2 \leq \frac{1}{d_{\max}L} =: R$.

**Proof of (60):** By the definition of $\sigma^2 := \left\|\sum_{\mathcal{I}\in\mathcal{E}^{(M)}} \sum_{\ell=1}^{L} \mathbb{E}\left[\left(\tilde{\Delta}_{\mathcal{I}}^{(\ell)}\right)^2\right]\right\|_2$, we get

$$\sigma^2 \overset{(a)}{=} \max_{\|u\|_2^2 + \|v\|_2^2 = 1} \left| \begin{bmatrix} u^T & v^T \end{bmatrix} \begin{bmatrix} \sum_{\mathcal{I}\in\mathcal{E}^{(M)}} \sum_{\ell=1}^{L} \mathbb{E}\left[\Delta_{\mathcal{I}}^{(\ell)}\left(\Delta_{\mathcal{I}}^{(\ell)}\right)^T\right] & 0 \\ 0 & \sum_{\mathcal{I}\in\mathcal{E}^{(M)}} \sum_{\ell=1}^{L} \mathbb{E}\left[\left(\Delta_{\mathcal{I}}^{(\ell)}\right)^T \Delta_{\mathcal{I}}^{(\ell)}\right] \end{bmatrix} \begin{bmatrix} u \\ v \end{bmatrix} \right| \tag{66}$$

$$\overset{(b)}{=} \max_{\|u\|_2^2 + \|v\|_2^2 = 1} \left\| \sum_{\mathcal{I}\in\mathcal{E}^{(M)}} \sum_{\ell=1}^{L} \mathbb{E}\left[\Delta_{\mathcal{I}}^{(\ell)}\left(\Delta_{\mathcal{I}}^{(\ell)}\right)^T\right] \right\|_2 \|u\|_2^2 + \left\| \sum_{\mathcal{I}\in\mathcal{E}^{(M)}} \sum_{\ell=1}^{L} \mathbb{E}\left[\left(\Delta_{\mathcal{I}}^{(\ell)}\right)^T \Delta_{\mathcal{I}}^{(\ell)}\right] \right\|_2 \|v\|_2^2 \tag{67}$$

$$= \max \left\{ \left\| \sum_{\mathcal{I}\in\mathcal{E}^{(M)}} \sum_{\ell=1}^{L} \mathbb{E}\left[\Delta_{\mathcal{I}}^{(\ell)}\left(\Delta_{\mathcal{I}}^{(\ell)}\right)^T\right] \right\|_2, \left\| \sum_{\mathcal{I}\in\mathcal{E}^{(M)}} \sum_{\ell=1}^{L} \mathbb{E}\left[\left(\Delta_{\mathcal{I}}^{(\ell)}\right)^T \Delta_{\mathcal{I}}^{(\ell)}\right] \right\|_2 \right\}, \tag{68}$$

where $(a)$ follows by the Courant-Fischer theorem where $u, v \in \mathbb{R}^n$; $(b)$ follows by the definition of $\|\cdot\|_2$.

Using the matrix version of Hölder's inequality in Appendix 5.4, we get

$$\left\| \sum_{\mathcal{I} \in \mathcal{E}^{(M)}} \sum_{\ell=1}^{L} \mathbb{E}\left[ \Delta_{\mathcal{I}}^{(\ell)} \left( \Delta_{\mathcal{I}}^{(\ell)} \right)^T \right] \right\|_2 \leq \sqrt{ \left\| \sum_{\mathcal{I} \in \mathcal{E}^{(M)}} \sum_{\ell=1}^{L} \mathbb{E}\left[ \Delta_{\mathcal{I}}^{(\ell)} \left( \Delta_{\mathcal{I}}^{(\ell)} \right)^T \right] \right\|_1 \left\| \sum_{\mathcal{I} \in \mathcal{E}^{(M)}} \sum_{\ell=1}^{L} \mathbb{E}\left[ \Delta_{\mathcal{I}}^{(\ell)} \left( \Delta_{\mathcal{I}}^{(\ell)} \right)^T \right] \right\|_\infty } \tag{69}$$

$$\overset{(a)}{=} \sqrt{ \max_{1 \leq i \leq n} \left( \sum_{j=1}^{n} \left| \sum_{\mathcal{I} \in \mathcal{E}^{(M)}} \sum_{\ell=1}^{L} \mathbb{E}\left[ \Delta_{\mathcal{I}}^{(\ell)} \left( \Delta_{\mathcal{I}}^{(\ell)} \right)^T \right] \right|_{ij} \right) \max_{1 \leq j \leq n} \left( \sum_{i=1}^{n} \left| \sum_{\mathcal{I} \in \mathcal{E}^{(M)}} \sum_{\ell=1}^{L} \mathbb{E}\left[ \Delta_{\mathcal{I}}^{(\ell)} \left( \Delta_{\mathcal{I}}^{(\ell)} \right)^T \right] \right|_{ij} \right) } \tag{70}$$

$$\leq \sqrt{ \left( \sum_{\mathcal{I} \in \mathcal{E}^{(M)}} \sum_{\ell=1}^{L} \max_{1 \leq i \leq n} \left( \sum_{j=1}^{n} \left| \mathbb{E}\left[ \Delta_{\mathcal{I}}^{(\ell)} \left( \Delta_{\mathcal{I}}^{(\ell)} \right)^T \right] \right|_{ij} \right) \right) \left( \sum_{\mathcal{I} \in \mathcal{E}^{(M)}} \sum_{\ell=1}^{L} \max_{1 \leq j \leq n} \left( \sum_{i=1}^{n} \left| \mathbb{E}\left[ \Delta_{\mathcal{I}}^{(\ell)} \left( \Delta_{\mathcal{I}}^{(\ell)} \right)^T \right] \right|_{ij} \right) \right) } \tag{71}$$

$$\overset{(b)}{\leq} 3 d_{\max} L \max_{i,j} \left| \mathbb{E}\left[ \left( \Delta_{\mathcal{I}}^{(\ell)} \right) (\Delta_{\mathcal{I}}^{(\ell)})^T) \right]_{ij} \right|, \tag{72}$$

where $(a)$ follows by the definitions of $\|\cdot\|_1$ and $\|\cdot\|_\infty$; $(b)$ follows by the fact that for each row in $\mathbb{E}\left[ \Delta_{\mathcal{I}}^{(\ell)} \left( \Delta_{\mathcal{I}}^{(\ell)} \right)^T \right]$, there are only three non-zero entries. The $(i,j)$-entry of matrix $\mathbb{E}\left[ \Delta_{\mathcal{I}}^{(\ell)} \left( \Delta_{\mathcal{I}}^{(\ell)} \right)^T \right]$ is the expectation of the inner product of the $i^{\text{th}}$ and $j^{\text{th}}$ rows of $\Delta_{\mathcal{I}}^{(\ell)}$. For each $i$, the $(i,j)$-entry is non-zero for $j$ such that $j = i$ or the $i^{\text{th}}$ element of the $j^{\text{th}}$ row of $\Delta_{\mathcal{I}}^{(\ell)}$ is non-zero. Also, there are two rows of $\Delta_{\mathcal{I}}^{(\ell)}$ whose $i^{\text{th}}$ entry is non-zero because for $\{i,j\} \in \phi(\mathcal{I})$, the $i^{\text{th}}$ element of the $j^{\text{th}}$ row of $\mathbb{E}\left[ \Delta_{\mathcal{I}}^{(\ell)} \left( \Delta_{\mathcal{I}}^{(\ell)} \right)^T \right]$ is non-zero, and there are two $j$'s such that $\{i,j\} \in \phi(\mathcal{I})$ for each $i$.

Let us further obtain a bound of (72). For $\{i,j\} \in \phi(\mathcal{I})$, by the definition of $\Delta_{\mathcal{I}}^{(\ell)}$, we get

$$\left| \mathbb{E}\left[ \left( \Delta_{\mathcal{I}}^{(\ell)} (\Delta_{\mathcal{I}}^{(\ell)})^T \right)_{ij} \right] \right| \leq \sum_k \left| \mathbb{E}\left[ \left( \Delta_{\mathcal{I}}^{(\ell)} \right)_{ik} (\Delta_{\mathcal{I}}^{(\ell)})_{jk} \right] \right|$$

$$= \frac{1}{4L^2 d_{\max}^2} \sum_k \left| \mathbb{E}\left[ \left( y_{ik,\mathcal{I}} - \frac{w_i}{w_i + w_k} \right) \left( y_{jk,\mathcal{I}} - \frac{w_j}{w_j + w_k} \right) \right] \right| \mathbb{I}\left[ \{i,k\}, \{j,k\} \in \phi(\mathcal{I}) \right] \tag{73}$$

$$\overset{(a)}{=} \frac{1}{4L^2 d_{\max}^2} \sum_k \left| \mathbb{E}\left[ \left( y_{ki,\mathcal{I}} - \frac{w_k}{w_i + w_k} \right) \left( y_{kj,\mathcal{I}} - \frac{w_k}{w_j + w_k} \right) \right] \right| \mathbb{I}\left[ \{i,k\}, \{j,k\} \in \phi(\mathcal{I}) \right] \tag{74}$$

$$= \frac{1}{4L^2 d_{\max}^2} \sum_k \left| \mathbb{E}\left[ y_{ki,\mathcal{I}} y_{kj,\mathcal{I}} - \frac{w_k}{w_i + w_k} y_{kj,\mathcal{I}} - \frac{w_k}{w_j + w_k} y_{ki,\mathcal{I}} + \frac{w_k}{w_i + w_k} \frac{w_k}{w_j + w_k} \right] \right| \mathbb{I}\left[ \{i,k\}, \{j,k\} \in \phi(\mathcal{I}) \right] \tag{75}$$

$$\overset{(b)}{\leq} \frac{4}{4L^2 d_{\max}^2} \sum_k \mathbb{I}\left[ \{i,k\}, \{j,k\} \in \phi(\mathcal{I}) \right] \overset{(c)}{\leq} \frac{8}{4L^2 d_{\max}^2} \leq \frac{2}{L^2 d_{\max}^2}, \tag{76}$$

where $(a)$ follows by the assumption that $y_{ik,\mathcal{I}} = 1 - y_{ki,\mathcal{I}}$; $(b)$ follows by the fact that $|y_{ki,\mathcal{I}}| \leq 1$ and $|y_{kj,\mathcal{I}}| \leq 1$; $(c)$ follows by the fact for $i \neq j$, there is only one item adjacent to both items $i$ and $j$, and for $i = j$, there are two. Applying (76) to (72), we get the bound $\left\| \sum_{\mathcal{I} \in \mathcal{E}^{(M)}} \sum_{\ell=1}^{L} \mathbb{E}\left[ \Delta_{\mathcal{I}}^{(\ell)} \left( \Delta_{\mathcal{I}}^{(\ell)} \right)^T \right] \right\| \leq \frac{6}{d_{\max} L}$.

Similarly, we can show that $\left\| \sum_{\mathcal{I} \in \mathcal{E}^{(M)}} \sum_{\ell=1}^{L} \mathbb{E}\left[ \left( \Delta_{\mathcal{I}}^{(\ell)} \right)^T \Delta_{\mathcal{I}}^{(\ell)} \right] \right\|$ is also bounded by $\frac{6}{d_{\max} L}$. Therefore, we get the desired bound $\sigma^2 \leq \frac{6}{L d_{\max}}$ from (68).

## 3.2 Proof of Lemma 6

**Outline:** To obtain a lower bound of the spectral gap $h(P)$ of reversible matrix $P$, we perform two tasks.

First, we introduce another reversible matrix $Q$ and establish a relation between $h(P)$ and $h(Q)$ in (77). We construct $Q$ so as to make it have certain conditions, which help us compute its spectral gap $h(Q)$ easily. The relation (77) shows that $h(P)$ is lowered-bounded by the multiplication of $h(Q)$ and some scaling factor. The derivation follows the proof of Lemma 6 in (Negahban, Oh and Shah 2016).

Second, we compute lower bounds for the scaling factor and $h(Q)$ in (79) and (80) respectively. To obtain a lower bound for the scaling in (79) is straightforward. To obtain a lower bound for $h(Q)$ needs some extra efforts. The spectral gap $h(Q)$ is defined as $1 - \lambda_{\max}(Q)$. Thus, an upper bound of $\lambda_{\max}(Q)$ can lead to a lower bound of $h(Q)$. We show that $\lambda_{\max}(Q)$ can be decomposed into three terms ((88), (89) and (90)) for each of which we derive an upper bound. To obtain (88) and (89), we use the matrix version of Hölder's inequality, and to obtain (90), we use the matrix Bernstein inequality. As in Lemma 5, to apply the matrix Bernstein inequality, we first compute two parameters in (97) and (98) and use them in obtaining (90).

---

**Proof dependencies:**
Lemma 6 ⟵ Equation (79), Equation (80)
   Equation (79)
   Equation (80) ⟵ Equation (88), Equation (89), Equation (90)
      Equation (88) ⟵ Hölder's inequality (Appendix 5.4)
      Equation (89) ⟵ Hölder's inequality
      Equation (90) ⟵ Matrix Bernstein inequality (Appendix 5.5)
         Matrix Bernstein inequality ⟵ Parameters (97) and (98)
            Parameter (97) ⟵ Hölder's inequality
            Parameter (98) ⟵ Hölder's inequality

---

**Proof:** Referring to Lemma 6 in (Negahban, Oh and Shah 2016), we can obtain the following lower bound on $h(P)$ using another reversible matrix $Q$:

$$h(P) \geq h(Q)\frac{\alpha}{\beta}, \tag{77}$$

where $\alpha := \min_{(i,j)} \left( \frac{w_i P_{ji}}{u_i Q_{ji}} \right)$, $\beta := \max_i \left( \frac{w_i}{u_i} \right)$, and $\boldsymbol{u}$ is the first eigenvector of $Q$. To obtain a lower bound on $h(P)$, we need to find $h(Q)$, $\alpha$, and $\beta$.

First, let us specify $Q$. $Q$ is the reversible transition matrix of random walks, which is defined as

$$Q_{ij} := \frac{1}{2d_j} \sum_{\mathcal{I}:\mathcal{I} \in \mathcal{E}^{(M)}} \mathbb{I}\left[ \{i,j\} \in \phi(\mathcal{I}) \right] = \frac{d_{ij}}{2d_j}, \tag{78}$$

where $d_{ij}$ is defined by the number of hyper-edges that have both item $i$ and item $j$.

From the reversible Markov chain $Q$, we can obtain the first eigenvector $\boldsymbol{u}$ of $Q$ by solving detailed balance equations: $u_j Q_{ij} = u_i Q_{ji}$. One can verify that $u_i = \frac{d_i}{\sum_{m=1}^n d_m}$ where $\sum_{m=1}^n d_m$.

To find a lower bound of $h(P)$ applying (77), let us assume that the following holds for now, of which we will provide proofs soon.

$$\frac{\alpha}{\beta} \geq \frac{1}{4b^2} \frac{d_{\min}}{d_{\max}}, \tag{79}$$

$$h(Q) \geq \frac{4}{90}. \tag{80}$$

Since $\frac{1}{2}d_{\mathrm{avg}} < d_i < \frac{3}{2}d_{\mathrm{avg}}$ holds with high probability by Lemma 7, we get

$$h(P) \geq \frac{1}{270b^2}. \tag{81}$$

This finishes the proof of (45).

### 3.2.1 Bound on $\alpha/\beta$

We obtain a lower bound of $\frac{\alpha}{\beta}$ by getting a lower bound of $\alpha$ and an upper bound of $\beta$. First, by the definition of $\alpha$, we get

$$\alpha = \min_{(i,j)} \left( \frac{w_i P_{ji}}{u_i Q_{ji}} \right) \overset{(a)}{=} \min_{(i,j)} \left( \frac{w_i \frac{d_{ij}}{2d_{\max}} \frac{w_i}{w_i + w_j}}{\frac{d_i}{\sum_{m=1}^{n} d_m} \frac{d_{ij}}{2d_i}} \right) \overset{(b)}{\geq} \frac{w_{\min}^2}{2w_{\max}} \frac{\sum_m d_m}{d_{\max}}, \tag{82}$$

where $(a)$ follows by the definitions of $P$ and $Q$; $(b)$ follows by the fact that $\frac{w_i}{w_i + w_j} \geq \frac{w_{\min}}{2w_{\max}}$, and $w_i \geq w_{\min}$.

Now, an upper bound of $\beta$ ends the proof. By the definition of $\beta$, we get

$$\beta = \max_i \left( \frac{w_i}{u_i} \right) \overset{(a)}{\leq} \frac{w_{\max}}{\frac{d_{\min}}{2\sum_m d_m}} = \frac{2w_{\max} \sum_m d_m}{d_{\min}}, \tag{83}$$

where $(a)$ follows by the fact that $w_i \leq w_{\max}$ and the fact that $d_i \geq d_{\min}$.

Using (82) and (83), we finally obtain

$$\frac{\alpha}{\beta} \geq \frac{1}{4b^2} \frac{d_{\min}}{d_{\max}}. \tag{84}$$

where the last inequality follows by the fact that $d_i$ concentrates around those expectation with high probability. Specifically, $\frac{1}{2}\binom{n-1}{M-1}p \leq d_i \leq \frac{3}{2}\binom{n-1}{M-1}p$ can be proved by Lemma 7.

### 3.2.2 Bound on $h(Q)$

The spectral gap of matrix $Q$ is defined as $h(Q) = 1 - \lambda_{\max}(Q)$ where $\lambda_{\max}(Q)$ is the second largest eigenvalue of $Q$. If matrix $Q$ is symmetric, we can obtain the spectral gap by subtracting the rank-1 projection matrix of the first eigenvector of $Q$ from original matrix $Q$, and getting the first eigenvalue of the subtracted matrix. However, $Q$ is neither symmetric nor easy to deal with.

Fortunately, we can find a symmetric matrix $S$ whose eigenvalues are the same as those of $Q$. The symmetric matrix can be expressed as

$$S = U^{-1/2} Q U^{1/2}, \tag{85}$$

where $U$ is a diagonal matrix such that $U_{ii} = u_i \left( \sum_m d_m \right) = d_i$.

As mentioned, we can compute $\lambda_{\max}(Q)$, the second largest eigenvalue of $Q$ in an alternative way: subtract the rank-1 projection matrix of $S$ from $S$ and get the first eigenvalue of the subtracted matrix. One can verify that the first eigenvector of $S$ is $u^{1/2} = \left( u_1^{1/2}, u_2^{1/2}, \cdots, u_n^{1/2} \right)^T$. Before computing $\lambda_{\max}(Q)$, we define a matrix $A$ such that $A_{ij} = d_{ij}$ for simplicity of analysis that will follow. Notice that $Q = \frac{1}{2}AU^{-1}$ holds. Now, let us begin to compute $\lambda_{\max}(Q)$.

$$\lambda_{\max}(Q) = \lambda_{\max}(S) = \left\| U^{-1/2} Q U^{1/2} - u_1^{1/2} \left( u_1^{1/2} \right)^T \right\|_2 = \left\| \frac{1}{2} U^{-1/2} A U^{-1/2} - u_1^{1/2} \left( u_1^{1/2} \right)^T \right\|_2 \tag{86}$$

$$\leq \left\| \frac{1}{2} U^{-1/2} A U^{-1/2} - \frac{1}{2d_{\text{avg}}} A \right\|_2 + \left\| \frac{1}{2d_{\text{avg}}} A - \mathbb{E}\left[ \frac{1}{2d_{\text{avg}}} A \right] \right\|_2 + \left\| \mathbb{E}\left[ \frac{1}{2d_{\text{avg}}} A \right] - u_1^{1/2} \left( u_1^{1/2} \right)^T \right\|_2, \tag{87}$$

where $d_{\text{avg}} := \mathbb{E}[d_i]$. Since $h(Q) = 1 - \lambda_{\max}(Q)$, upper bounds of the three terms lead to a lower bound of $h(Q)$. Soon, we will show that the following three bounds for some range of $p$.

$$\left\| \frac{1}{2} U^{-1/2} A U^{-1/2} - \frac{1}{2d_{\text{avg}}} A \right\|_2 \leq \frac{11}{90}, \tag{88}$$

$$\left\| \mathbb{E}\left[\frac{1}{2d_{\mathrm{avg}}}A\right] - u_1^{1/2}\left(u_1^{1/2}\right)^T \right\|_2 \le \frac{13}{18}, \tag{89}$$

$$\left\| \frac{1}{2d_{\mathrm{avg}}}A - \mathbb{E}\left[\frac{1}{2d_{\mathrm{avg}}}A\right] \right\|_2 \le \frac{1}{9}. \tag{90}$$

Then all these bounds give us that $h(Q) \ge \frac{2}{45}$. Now, let us provide the proofs of (88), (89) and (90).

**Proof of (88):** Using the matrix version of Hölder's inequality, we get

$$\frac{1}{2}\left\| U^{-1/2}AU^{-1/2} - \frac{A}{d_{\mathrm{avg}}} \right\|_2 \le \frac{1}{2}\left\| U^{-1/2}AU^{-1/2} - \frac{A}{d_{\mathrm{avg}}} \right\|_1 \left\| U^{-1/2}AU^{-1/2} - \frac{A}{d_{\mathrm{avg}}} \right\|_\infty \overset{(a)}{\le} \frac{1}{2}\max_i\left\{ \sum_j \left| \frac{d_{ij}}{\sqrt{d_id_j}} - \frac{d_{ij}}{d_{\mathrm{avg}}} \right| \right\} \tag{91}$$

$$\overset{(b)}{\le} \frac{1}{2}\max_i\left\{ \sum_j d_{ij}\max\left\{ \left| \frac{1}{d_{\min}} - \frac{1}{d_{\mathrm{avg}}} \right|, \left| \frac{1}{d_{\max}} - \frac{1}{d_{\mathrm{avg}}} \right| \right\} \right\} = \frac{1}{2}\max_i\left\{ 2d_i\max\left\{ \left| \frac{1}{d_{\min}} - \frac{1}{d_{\mathrm{avg}}} \right|, \left| \frac{1}{d_{\max}} - \frac{1}{d_{\mathrm{avg}}} \right| \right\} \right\} \tag{92}$$

$$\le d_{\max}\cdot\max\left\{ \left| \frac{1}{d_{\min}} - \frac{1}{d_{\mathrm{avg}}} \right|, \left| \frac{1}{d_{\max}} - \frac{1}{d_{\mathrm{avg}}} \right| \right\} \overset{(c)}{\le} \frac{11}{90}, \tag{93}$$

where $(a)$ follows by the definitions of $\|\cdot\|_1$ and $\|\cdot\|_\infty$, and by the fact that $\left(U^{-1/2}AU^{-1/2}\right)_{ij} = d_{ij}/\sqrt{d_id_j}$ and $A_{ij} = d_{ij}$; $(b)$ follows by the fact that $\frac{1}{d_{\max}} \le \frac{1}{\sqrt{d_id_j}} \le \frac{1}{d_{\min}}$; $(c)$ follows by Lemma 7 which states $\frac{9}{10}d_{\mathrm{avg}} \le d_i \le \frac{11}{10}d_{\mathrm{avg}}$ with high probability.

**Proof of (89):** Similarly, using the matrix version of Hölder's inequality and the definitions of $\|\cdot\|_1$ and $\|\cdot\|_\infty$, we get

$$\left\| \mathbb{E}\left[\frac{1}{2d_{\mathrm{avg}}}A\right] - u_1^{1/2}\left(u_1^{1/2}\right)^T \right\|_2 \le \max_i\left\{ \sum_j \left| \frac{\mathbb{E}[d_{ij}]}{2d_{\mathrm{avg}}} - \frac{\sqrt{d_id_j}}{\sum_m d_m} \right| \right\} \overset{(a)}{=} \max_i\left\{ \sum_j \left| \frac{1}{2d_{\mathrm{avg}}}\binom{n-2}{M-2}\frac{p}{M-1} - \frac{\sqrt{d_id_j}}{\sum_m d_m} \right| \right\} \tag{94}$$

$$\overset{(b)}{=} \max_i\left\{ \sum_j \left| \frac{1}{2(n-1)} - \frac{\sqrt{d_id_j}}{\sum_m d_m} \right| \right\} \le n\cdot\max\left\{ \left| \frac{1}{2(n-1)} - \frac{d_{\min}}{nd_{\max}} \right|, \left| \frac{1}{2(n-1)} - \frac{d_{\max}}{nd_{\min}} \right| \right\} \overset{(c)}{\le} \frac{13}{18}, \tag{95}$$

where $(a)$ follows by the fact that $\mathbb{E}[d_{ij}] = \sum_{\mathcal{I}:i,j\in\mathcal{I}}\mathbb{E}\left[(A_{\mathcal{I}})_{ij}\right] = \binom{n-2}{M-2}\frac{p}{M-1}$; $(b)$ follows by the facts that $\binom{n-2}{M-2} = \binom{n-1}{M-1}\frac{M-1}{n-1}$ and $\binom{n-1}{M-1}p = d_{\mathrm{avg}}$; $(c)$ follows by Lemma 7 which states $\frac{9}{10}d_{\mathrm{avg}} \le d_i \le \frac{11}{10}d_{\mathrm{avg}}$ with high probability.

**Proof of (90):** We prove (90) applying the matrix Bernstein inequality. First, let us express $A$ as the summation of independent random matrices.

$$A = \sum_{\mathcal{I}} A_{\mathcal{I}}, \quad (A_{\mathcal{I}})_{ij} := \mathbb{I}\left[\mathcal{I} \in \mathcal{E}^{(M)}, \{i,j\} \in \phi(\mathcal{I})\right] = \mathbb{I}\left[\mathcal{I} \in \mathcal{E}^{(M)}\right]\mathbb{I}\left[\{i,j\} \in \phi(\mathcal{I})\right], \tag{96}$$

where $\mathcal{I}$ is a hyper-edge of the set $[n]$.

Using the matrix Bernstein inequality, when we assume the following two conditions hold,

$$R = \|A_{\mathcal{I}} - \mathbb{E}[A_{\mathcal{I}}]\|_2 \le 3, \tag{97}$$

$$\sigma^2 = \left\| \sum_{\mathcal{I}} \mathbb{E}\left[(A_{\mathcal{I}} - \mathbb{E}[A_{\mathcal{I}}])^2\right] \right\|_2 \le 3d_{\mathrm{avg}}, \tag{98}$$

then we can show that

$$\Pr\left[\|A - \mathbb{E}[A]\|_2 \geq t\right] \leq 2n \exp\left(\frac{-t^2/2}{\sigma^2 + Rt}\right) \leq 2n \exp\left(\frac{-t^2/2}{3d_{\mathrm{avg}} + 3t}\right) \overset{(a)}{=} 2n \exp\left(\frac{-16 \log n}{3 + 12\sqrt{\frac{\log n}{d_{\mathrm{avg}}}}}\right) \leq 2n^{-1/15}, \quad (99)$$

where $(a)$ follows by setting $t = 4\sqrt{d_{\mathrm{avg}} \log n}$. From this, we get $\left\|\frac{1}{2d_{\mathrm{avg}}}A - \mathbb{E}\left[\frac{1}{2d_{\mathrm{avg}}}A\right]\right\|_2 \leq 2\sqrt{\frac{\log n}{d_{\mathrm{avg}}}}$. Also, for $p > 324\frac{\log n}{\binom{n-1}{M-1}}$, since $d_{\mathrm{avg}} = \binom{n-1}{M-1}p = 324 \log n$ holds, we finished the proof of (90). Now, we provide the proofs of (97) and (98).

*Proof of (97):* $R$ can be bounded as follows.

$$R \overset{(a)}{\leq} \|A_{\mathcal{I}} - \mathbb{E}[A_{\mathcal{I}}]\|_1 \|A_{\mathcal{I}} - \mathbb{E}[A_{\mathcal{I}}]\|_\infty \overset{(b)}{\leq} \max_i \left\{\sum_j \left|(A_{\mathcal{I}} - \mathbb{E}[A_{\mathcal{I}}])_{ij}\right|\right\} \overset{(c)}{\leq} 2\left|1 - \frac{p}{M-1}\right| + (M-2)\frac{p}{M-1} \leq 3, \quad (100)$$

where $(a)$ follows by the matrix version of Hölder's inequality; $(b)$ follows by the definitions of $\|\cdot\|_1$ and $\|\cdot\|_\infty$; $(c)$ follows by the fact that since there are two adjacent items for each item, $(A_{\mathcal{I}})_{ij} = 1$ at two elements for each row, and $\mathbb{E}\left[(A_{\mathcal{I}})_{ij}\right] = \frac{p}{M-1}$ by the fact that $(A_{\mathcal{I}})_{ij} \sim \mathrm{Bern}\left(\frac{p}{M-1}\right)$. We can derive the distribution $(A_{\mathcal{I}})_{ij} \sim \mathrm{Bern}\left(\frac{p}{M-1}\right)$ as follows.

By the definition of (96), we see that $(A_{\mathcal{I}})_{ij} = \mathbb{I}\left[\mathcal{I} \in \mathcal{E}^{(M)}\right] \mathbb{I}\left[\{i,j\} \in \phi(\mathcal{I})\right]$. We know that $\mathbb{I}\left[\mathcal{I} \in \mathcal{E}^{(M)}\right] \sim \mathrm{Bern}(p)$ by the assumption that every hyper-edge is chosen independently with probability $p$. Also, since a circular permutation for each $\mathcal{I}$ is chosen uniformly at random, we can compute $\Pr\left[\{i,j\} \in \phi(\mathcal{I})\right]$ as follows.

$$\Pr\left[\{i,j\} \in \phi(\mathcal{I})\right] = \frac{\text{\# of circular permutations in which items } i \text{ and } j \text{ are adjacent}}{\text{\# of circular permutations}} = \frac{(M-2)!}{(M-1)!} = \frac{1}{M-1}. \quad (101)$$

Hence, we get $(A_{\mathcal{I}})_{ij} \sim \mathrm{Bern}\left(\frac{p}{M-1}\right)$ for $i, j \in \mathcal{I}$. This ends the proof of (97).

*Proof of (98):* For notational simplicity, we let $A^* := \sum_{\mathcal{I}} \mathbb{E}\left[(A_{\mathcal{I}} - \mathbb{E}[A_{\mathcal{I}}])^2\right]$. Similarly, using the matrix version of Hölder's inequality and the definitions of $\|\cdot\|_1$ and $\|\cdot\|_\infty$, we get

$$\sigma^2 \leq \max_i \left\{\sum_j \left|A^*_{ij}\right|\right\}. \quad (102)$$

Now let us obtain an upper bound of $A^*_{ij}$. First, for the case of $i \neq j$,

$$\left|A^*_{ij}\right| = \left|\sum_{\mathcal{I}: i,j \in \mathcal{I}} \sum_{k \neq i,j} \mathbb{E}\left[(A_{\mathcal{I}} - \mathbb{E}[A_{\mathcal{I}}])_{ik}(A_{\mathcal{I}} - \mathbb{E}[A_{\mathcal{I}}])_{kj}\right]\right| \leq \sum_{\mathcal{I}: i,j \in \mathcal{I}} \sum_{k \neq i,j} \left|\mathbb{E}\left[(A_{\mathcal{I}})_{ik}(A_{\mathcal{I}})_{kj}\right] - \mathbb{E}\left[(A_{\mathcal{I}})_{ik}\right] \mathbb{E}\left[(A_{\mathcal{I}})_{kj}\right]\right| \quad (103)$$

$$\overset{(a)}{=} \sum_{\mathcal{I}: i,j \in \mathcal{I}} \sum_{k \neq i,j} \left|\frac{p}{(M-1)(M-2)} - \frac{p}{(M-1)^2}\right| = \binom{n-2}{M-2}(M-2)\left|\frac{p}{(M-1)(M-2)} - \frac{p}{(M-1)^2}\right| \quad (104)$$

$$= \binom{n-1}{M-1}\frac{M-1}{n-1}(M-2)\frac{p}{(M-1)^2(M-2)} \leq \frac{2}{n-1}\binom{n-1}{M-1}p = \frac{2}{n-1}d_{\mathrm{avg}}, \quad (105)$$

where $(a)$ follows by the fact that $(A_{\mathcal{I}})_{ik}(A_{\mathcal{I}})_{kj} \sim \mathrm{Bern}\left(\frac{p}{(M-1)(M-2)}\right)$. $(A_{\mathcal{I}})_{ik}(A_{\mathcal{I}})_{kj}$ is equal to 1, when we have $i - k - j$ in the formed circular permutation. As previously shown, we can derive the distribution $(A_{\mathcal{I}})_{ik}(A_{\mathcal{I}})_{kj} = 1$ as follows.

$$\Pr\left[(A_{\mathcal{I}})_{ik}(A_{\mathcal{I}})_{kj} = 1\right] = \frac{\text{\# of circular permutations in which items } i \text{ and } j \text{ are adjacent to item } k}{\text{\# of circular permutations}} \quad (106)$$

$$= \frac{(M-3)!}{(M-1)!} = \frac{1}{(M-1)(M-2)}. \tag{107}$$

For the case of $i = j$, we get

$$|A_{ii}^*| = \left| \sum_{\mathcal{I}:i\in\mathcal{I}} \sum_{k\neq i} \mathbb{E}\left[ (A_{\mathcal{I}} - \mathbb{E}\left[A_{\mathcal{I}}\right])_{ik} (A_{\mathcal{I}} - \mathbb{E}\left[A_{\mathcal{I}}\right])_{ki} \right] \right| \leq \sum_{\mathcal{I}:i\in\mathcal{I}} \sum_{k\neq i} \left| \mathbb{E}\left[ (A_{\mathcal{I}})_{ik}^2 \right] - \mathbb{E}\left[ (A_{\mathcal{I}})_{ik} \right]^2 \right| \tag{108}$$

$$= \binom{n-1}{M-1}(M-1) \left| \frac{p}{M-1} - \frac{p^2}{(M-1)^2} \right| \leq \binom{n-1}{M-1} p = d_{\text{avg}}. \tag{109}$$

Applying (109) and (105) to (102) ends the proof of (98).

$$\sigma^2 \leq 3d_{\text{avg}}. \tag{110}$$

# 4    Proof of Theorem 1

**Theorem 1.** *Fix $\epsilon \in (0, \frac{1}{2})$. Given an $M$-wise comparison graph $\mathcal{G} = ([n], \mathcal{E}^{(M)})$, if*

$$\binom{n}{M} pL \lesssim (1-\epsilon) \frac{n \log n}{\Delta_K^2} \frac{1}{M}, \tag{111}$$

*then for any ranking scheme $\psi$, there exists a preference score vector $\boldsymbol{w}$ with seperation measure $\Delta_K$ such that $P_e(\psi) \geq \epsilon$.*

**Outline:** Overall, the proof to be presented follows the line of steps in the proof of Theorem 2 in (Chen and Suh 2015). Similarly as in (Chen and Suh 2015), we intend to bound the minimax probability of error to characterize the conditions under which the probability cannot be made arbitrarily close to zero, using a generalized version of Fano's inequality (Han and Verdú 1994). However, the details of the steps are more involved including combinatorial calculations, as we consider a more general model.

We first construct a set of hypotheses, and impose a uniform distribution over them. We then apply the generalized Fano's inequality to obtain a lower bound on the probability of error. This lets us able to identify conditions under which the probability of error cannot be made arbitrarily zero.

At the end of the process, we obtain a sum of Kullback-Leibler (KL) divergences in (122). Computing its upper bound provides a lower bound of the probability of error, and it ends the proof. Depending on the hypotheses, the summand can be computed in four different ways. We divide-and-conquer and compute (122) in *Cases 1—4* and denote it by $D_1$, $D_2$, $D_3$ and $D_4$ respectively.

Finally, we show $D_4 = 0$, obtain an upper bound of $D_1 + D_2$ in (134) and that of $D_3$ in (135), and end the proof.

> **Proof dependencies:**
> Theorem 4 $\longleftarrow$ Equation (122)
>   Weighted sum of KL divergences $D_1$, $D_2$, $D_3$ and $D_4$ (Equation (122)) $\longleftarrow$ Equation (134), Equation (135)
>     $D_4 = 0$ (Equation (132))
>     Bound of $D_1 + D_2$ (Equation (134)) $\longleftarrow$ Equation (139), Equation (140)
>     Bound of $D_3$ (Equation (135)) $\longleftarrow$ Equation (162), Equation (163)

**Proof:** We construct a finite set of hypotheses $\mathcal{H}$ and carry out an analysis based on classical Fano-type arguments. Each hypothesis is represented by a permutation $\sigma_h \in \mathcal{H}$ over $[n]$ and we denote by $\sigma_h(i)$ and $\sigma_h([K])$ the index of the $i^{th}$ ranked item and the index set of all top-$K$ items respectively.

We choose a set of hypotheses and some prior to be imposed on them. Suppose that the values of $\boldsymbol{w}$ are fixed up to permutation in such a way that

$$\forall \sigma_h \in \mathcal{H}, \ w_{\sigma_h(i)} = \begin{cases} w_K & \text{if } 1 \leq i \leq K \\ w_{K+1} & \text{if } K < i \leq n, \end{cases} \tag{112}$$

where we abuse the notation $w_K$, $w_{K+1}$ to represent any two values satisfying

$$\frac{w_K - w_{K+1}}{w_{\max}} = \Delta_K > 0. \tag{113}$$

Additionally, we impose a uniform prior over a collection $\mathcal{H}$ of $|\mathcal{H}| = \max(K, n - K) + 1$ hypotheses regarding the permutation: if $K < \frac{n}{2}$, then

$$\forall \sigma_h \in \mathcal{H}, \ \mathbb{P}[\sigma_h] = \frac{1}{|\mathcal{H}|}, \ \sigma_h([K]) = \mathcal{K}_h, \ \text{for } \mathcal{K}_h = \{2, ..., K\} \cup \{h\}, \ (h = 1, K+1, ..., n), \tag{114}$$

and if $K \geq \frac{n}{2}$, then

$$\forall \sigma_h \in \mathcal{H}, \ \mathbb{P}[\sigma_h] = \frac{1}{|\mathcal{H}|}, \ \sigma_h([K]) = \mathcal{K}_h, \ \text{for } \mathcal{K}_h = \{1, ..., K+1\} \backslash \{h\}, \ (h = 1, ..., K+1). \tag{115}$$

Note that $|\mathcal{H}| \geq \frac{n}{2}$.

In words, each alternative hypothesis is made by interchanging two indices of the hypothesis complying to $\sigma_h([K]) = [K]$. Denoting by $P_{e,\mathcal{H}}$ the average probability of error with respect to the constructed prior, one can verify the minimax probability of error $P_e$ to be at least $P_{e,\mathcal{H}}$.

Let us begin our proof that modifies the arguments in (Chen and Suh 2015) for the model of our interest. To take partial $M$-wise observations into account, we introduce an erased version of $s_{\mathcal{I}} := (s_{\mathcal{I}}^{(1)}, s_{\mathcal{I}}^{(2)}, \ldots, s_{\mathcal{I}}^{(L)})$ such that

$$z_{\mathcal{I}} = \begin{cases} s_{\mathcal{I}} & \text{w.p.} \quad p; \\ \text{erasure} & \text{otherwise.} \end{cases}, \tag{116}$$

where we denote by $\boldsymbol{Z} := \{z_{\mathcal{I}} : \text{for all possible } \mathcal{I}\text{'s}\}$ the collection of observed samples.

Then, applying the generalized Fano's inequality (Han and Verdu 1994), we get

$$P_e \geq 1 - \frac{1}{\log |\mathcal{H}|} \left\{ \frac{1}{|\mathcal{H}|^2} \sum_{\sigma_a, \sigma_b \in \mathcal{H}} D(P_{\boldsymbol{Z}|\sigma=\sigma_a} || P_{\boldsymbol{Z}|\sigma=\sigma_b}) + \log 2 \right\} \tag{117}$$

$$\overset{(a)}{=} 1 - \frac{1}{\log |\mathcal{H}|} \left\{ \frac{1}{|\mathcal{H}|^2} \sum_{\sigma_a, \sigma_b \in \mathcal{H}} \sum_{\mathcal{I}} D(P_{z_{\mathcal{I}}|\sigma=\sigma_a} || P_{z_{\mathcal{I}}|\sigma=\sigma_b}) + \log 2 \right\} \tag{118}$$

$$\overset{(b)}{=} 1 - \frac{1}{\log |\mathcal{H}|} \left\{ \frac{p}{|\mathcal{H}|^2} \sum_{\sigma_a, \sigma_b \in \mathcal{H}} \sum_{\mathcal{I}} D(P_{s_{\mathcal{I}}|\sigma=\sigma_a} || P_{s_{\mathcal{I}}|\sigma=\sigma_b}) + \log 2 \right\} \tag{119}$$

$$\overset{(c)}{=} 1 - \frac{1}{\log |\mathcal{H}|} \left\{ \frac{pL}{|\mathcal{H}|^2} \sum_{\sigma_a, \sigma_b \in \mathcal{H}} \sum_{\mathcal{I}} D(P_{s_{\mathcal{I}}^{(1)}|\sigma=\sigma_a} || P_{s_{\mathcal{I}}^{(1)}|\sigma=\sigma_b}) + \log 2 \right\} \tag{120}$$

$$\overset{(d)}{=} 1 - \frac{1}{\log |\mathcal{H}|} \left\{ pL \sum_{\mathcal{I}} D(P_{s_{\mathcal{I}}^{(1)}|\sigma=\sigma_1} || P_{s_{\mathcal{I}}^{(1)}|\sigma=\sigma_{K+1}}) + \log 2 \right\}, \tag{121}$$

where $(a)$ follows by the independence between two hyper-edges; $(b)$ follows by the distribution of $z_{\mathcal{I}}$; $(c)$ follows by the independence of $s_{\mathcal{I}}^{(\ell)}$ over $\ell$; $(d)$ follows by the fact that for any pair of hypotheses they differ by one item and this leads the summation over all possible $\mathcal{I}$'s to the same KL divergence.

To identify conditions under which $P_e$ cannot be made arbitrarily close to zero, meaning top-$K$ ranking is infeasible, we seek to obtain a lower bound on $P_e$. To that end, we derive an upper bound on $\sum_{\mathcal{I}} D(P_{s_{\mathcal{I}}^{(1)}|\sigma=\sigma_1} || P_{s_{\mathcal{I}}^{(1)}|\sigma=\sigma_{K+1}})$. It turns out that $\sum_{\mathcal{I}} D(P_{s_{\mathcal{I}}^{(1)}|\sigma=\sigma_1} || P_{s_{\mathcal{I}}^{(1)}|\sigma=\sigma_{K+1}})$ is upper-bounded by

$$\sum_{\mathcal{I}} D(P_{s_{\mathcal{I}}^{(1)}|\sigma=\sigma_1} || P_{s_{\mathcal{I}}^{(1)}|\sigma=\sigma_{K+1}}) \leq \binom{n}{M} \frac{M}{n} c_0 \Delta_K^2, \tag{122}$$

where $c_0$ is a numerical constant.

We will soon prove (122) in detail. For the time being, let us proceed to characterize a necessary condition for reliable top-$K$ ranking. Applying (122) to (121), we get

$$P_e \geq 1 - \frac{1}{\log |\mathcal{H}|} \left\{ pL \left( \binom{n}{M} \frac{M}{n} c_0 \Delta_K^2 \right) + \log 2 \right\} = 1 - \frac{1}{\log |\mathcal{H}|} \left\{ c_0 \binom{n}{M} pL \frac{M}{n} \Delta_K^2 + \log 2 \right\}. \tag{123}$$

Fix $\epsilon \in (0, \frac{1}{2})$. Then, $P_e > \epsilon$ if $c_0 \binom{n}{M} pL \frac{M}{n} \Delta_K^2 < (1-\epsilon) \log |\mathcal{H}| - \log 2$. From this, we can obtain a necessary condition for reliable top-$K$ ranking:

$$c_0 \binom{n}{M} pL \frac{M}{n} \Delta_K^2 \geq \log |\mathcal{H}| - \log 2 \iff \binom{n}{M} pL \geq \frac{n(\log(n/2) - \log 2)}{M \Delta_K^2} \frac{1}{c_0} \iff \binom{n}{M} pL \gtrsim \frac{n \log n}{M \Delta_K^2}. \tag{124}$$

We can see that this gives us the claimed result of (111). As shown above, a key step to identifying the necessary condition is to show (122).

## 4.1 Bound on $\sum_{\mathcal{I}} D(P_{s_{\mathcal{I}}^{(1)}|\sigma=\sigma_1} || P_{s_{\mathcal{I}}^{(1)}|\sigma=\sigma_{K+1}})$: Proof of (122)

Here, we will upper-bound a sum of the Kullback-Leibler (KL) divergences $\sum_{\mathcal{I}} D(P_{s_{\mathcal{I}}^{(1)}|\sigma=\sigma_1} || P_{s_{\mathcal{I}}^{(1)}|\sigma=\sigma_{K+1}})$. Notice that $\sigma_1 = \{1, 2, ..., K\}$ and $\sigma_{K+1} = \{2, ..., K, K+1\}$. To show this, we consider four different cases, for each of which an observation set $\mathcal{I}$ of $M$ items includes certain items as follows.

- *Case 1*: $\mathcal{I}$ includes item 1 and does not include item $K+1$; the number of such $\mathcal{I}$ is $\binom{n-2}{M-1}$.

- *Case 2*: $\mathcal{I}$ includes item $K+1$ and does not include item 1; the number of such $\mathcal{I}$ is also $\binom{n-2}{M-1}$.

- *Case 3*: $\mathcal{I}$ includes both items 1 and $K+1$; the number of such $\mathcal{I}$ is $\binom{n-2}{M-2}$.

- *Case 4*: $\mathcal{I}$ includes neither item 1 nor item $K+1$; the number of such $\mathcal{I}$ is $\binom{n-2}{M}$.

Note that the four exclusive cases form a partition of the set of all possible $\mathcal{I}$. Now, let us compute the KL divergence for each case.

*Case 1* : $1 \in \mathcal{I}$ and $K+1 \notin \mathcal{I}$:

Let $R$ be the rank of item 1 within the permutation $s_{\mathcal{I}}^{(1)}$. Given $\sigma = \sigma_1$, we can obtain the probability of $s_{\mathcal{I}}^{(1)} = (i_1, i_2, \ldots, i_R = 1, \ldots, i_M)$, which we denote by $p_{s_{\mathcal{I}}^{(1)}|\sigma_1}$, according to the PL model as follows.

$$p_{s_{\mathcal{I}}^{(1)}|\sigma_1} = \prod_{r=1}^{M} \frac{w_{i_r}}{\sum_{m=r}^{M} w_{i_m}} = \left( \prod_{r=1}^{R-1} \frac{w_{i_r}}{w_K + \sum_{m \in [r,M] \setminus \{R\}} w_{i_m}} \right) \frac{w_K}{w_K + \sum_{m=R+1}^{M} w_{i_m}} \left( \prod_{r=R+1}^{M} \frac{w_{i_r}}{\sum_{m=r}^{M} w_{i_m}} \right). \tag{125}$$

Given $\sigma = \sigma_{K+1}$, we can obtain the probability of $s_{\mathcal{I}}^{(1)} = (i_1, i_2, \ldots, i_R = 1, \ldots, i_M)$ by substituting $w_K$ with $w_{K+1}$, as item 1 is not among the top-$K$ ranked in $\sigma_{K+1}$.

$$p_{s_{\mathcal{I}}^{(1)}|\sigma_{K+1}} = \left( \prod_{r=1}^{R-1} \frac{w_{i_r}}{w_{K+1} + \sum_{m : m \in [r,M] \setminus \{R\}} w_{i_m}} \right) \frac{w_{K+1}}{w_{K+1} + \sum_{m=R+1}^{M} w_{i_m}} \left( \prod_{r=R+1}^{M} \frac{w_{i_r}}{\sum_{m=r}^{M} w_{i_m}} \right) \tag{126}$$

Computing the KL divergence for this case, which we will denote by $D_1$, we get

$$D_1 = D(P_{s_{\mathcal{I}}^{(1)}|\sigma=\sigma_1} || P_{s_{\mathcal{I}}^{(1)}|\sigma=\sigma_{K+1}}) = \sum_{R=1}^{M} \sum_{s_{\mathcal{I}}^{(1)} : i_R = 1} p_{s_{\mathcal{I}}^{(1)}|\sigma_1} \log \frac{p_{s_{\mathcal{I}}^{(1)}|\sigma_1}}{p_{s_{\mathcal{I}}^{(1)}|\sigma_{K+1}}}. \tag{127}$$

*Case 2* : $K+1 \in \mathcal{I}$ and $1 \notin \mathcal{I}$:

This case is similar to *Case 1* except that the roles of items 1 and $K+1$ are swapped. Thus, the probability of $s_{\mathcal{I}}^{(1)} = (i_1, i_2, \ldots, i_R = K+1, \ldots, i_M)$ given $\sigma = \sigma_1$ is equivalent to (126), and that of $s_{\mathcal{I}}^{(1)} = (i_1, i_2, \ldots, i_R = K+1, \ldots, i_M)$ given $\sigma = \sigma_{K+1}$ is equivalent to (125).

Computing the KL divergence for this case, which we will denote by $D_2$, we get

$$D_2 = D(P_{s_{\mathcal{I}}^{(1)}|\sigma=\sigma_1} || P_{s_{\mathcal{I}}^{(1)}|\sigma=\sigma_{K+1}}) = \sum_{R=1}^{M} \sum_{s_{\mathcal{I}}^{(1)}:i_R=1} p_{s_{\mathcal{I}}^{(1)}|\sigma_{K+1}} \log \frac{p_{s_{\mathcal{I}}^{(1)}|\sigma_{K+1}}}{p_{s_{\mathcal{I}}^{(1)}|\sigma_1}}. \tag{128}$$

*Case 3* : $1 \in \mathcal{I}$ and $K+1 \in \mathcal{I}$:

Let $R_1$ and $R_{K+1}$ be the ranks of items 1 and $K+1$ respectively within the permutation $s_{\mathcal{I}}^{(1)}$. Given $\sigma = \sigma_1$, we obtain the probability of $s_{\mathcal{I}}^{(1)} = (i_1, i_2, \ldots, i_{R_1-1}, i_{R_1} = 1, \ldots, i_{R_{K+1}-1}, i_{R_{K+1}} = K+1, \ldots, i_M)$ as follows.

$$p_{s_{\mathcal{I}}^{(1)}|\sigma_1} = \left( \prod_{r=1}^{R_1-1} \frac{w_{i_r}}{w_K + w_{K+1} + \sum_{m:m\in[r,M]\setminus\{R_1,R_{K+1}\}} w_{i_m}} \right) \left( \frac{w_K}{w_K + w_{K+1} \sum_{m:m\in[R_1+1,M]\setminus\{R_{K+1}\}} w_{i_m}} \right)$$
$$\times \left( \prod_{r=R_1+1}^{R_{K+1}-1} \frac{w_{i_r}}{w_{K+1} \sum_{m:m\in[r,M]\setminus\{R_{K+1}\}} w_{i_m}} \right) \left( \frac{w_{K+1}}{w_{K+1} + \sum_{m=R_{K+1}+1}^{M} w_{i_m}} \right) \left( \prod_{r=R_{K+1}+1}^{M} \frac{w_{i_r}}{\sum_{m=r}^{M} w_{i_m}} \right). \tag{129}$$

Similarly, given $\sigma = \sigma_{K+1}$, we get

$$p_{s_{\mathcal{I}}^{(1)}|\sigma_{K+1}} = \left( \prod_{r=1}^{R_1-1} \frac{w_{i_r}}{w_{K+1} + w_K + \sum_{m:m\in[r,M]\setminus\{R_1,R_{K+1}\}} w_{i_m}} \right) \left( \frac{w_{K+1}}{w_{K+1} + w_K + \sum_{m:m\in[R_1+1,M]\setminus\{R_{K+1}\}} w_{i_m}} \right)$$
$$\times \left( \prod_{r=R_1+1}^{R_{K+1}-1} \frac{w_{i_r}}{w_K \sum_{m:m\in[r,M]\setminus\{R_{K+1}\}} w_{i_m}} \right) \left( \frac{w_K}{w_K + \sum_{m=R_{K+1}+1}^{M} w_{i_m}} \right) \left( \prod_{r=R_{K+1}+1}^{M} \frac{w_{i_r}}{\sum_{m=r}^{M} w_{i_m}} \right). \tag{130}$$

In (129) and (130), we consider the case where item 1 is ranked higher than item $K+1$ is, namely $R_1 < R_{K+1}$. Let us consider the symmetric case by assuming $R_{K+1}$ is the rank of item 1 and $R_1$ is the rank of item $K+1$, where $R_1 < R_{K+1}$. In this case, we can simply swap $w_K$ and $w_{K+1}$ in (129) and (130). Thus, we can obtain the probability of $\tilde{s}_{\mathcal{I}}^{(1)} = (i_1, i_2, \ldots, i_{R_1-1}, i_{R_1} = K+1, \ldots, i_{R_{K+1}-1}, i_{R_{K+1}} = 1, \ldots, i_M)$ as $p_{\tilde{s}_{\mathcal{I}}^{(1)}|\sigma_1} = p_{s_{\mathcal{I}}^{(1)}|\sigma_{K+1}}$ and $p_{\tilde{s}_{\mathcal{I}}^{(1)}|\sigma_{K+1}} = p_{s_{\mathcal{I}}^{(1)}|\sigma_1}$. Using these facts, we can simplify the computation of the KL divergence for this case, which we denote by $D_3$, as follows.

$$D_3 = \sum_{R_1<R_{K+1}} \sum_{s_{\mathcal{I}}^{(1)}:i_{R_1}=1,i_{R_{K+1}}=K+1} p_{s_{\mathcal{I}}^{(1)}|\sigma_1} \log \frac{p_{s_{\mathcal{I}}^{(1)}|\sigma_1}}{p_{s_{\mathcal{I}}^{(1)}|\sigma_{K+1}}} + \sum_{R_1>R_{K+1}} \sum_{s_{\mathcal{I}}^{(1)}:i_{R_1}=1,i_{R_{K+1}}=K+1} p_{s_{\mathcal{I}}^{(1)}|\sigma_1} \log \frac{p_{s_{\mathcal{I}}^{(1)}|\sigma_1}}{p_{s_{\mathcal{I}}^{(1)}|\sigma_{K+1}}}$$

$$= \sum_{R_1<R_{K+1}} \sum_{s_{\mathcal{I}}^{(1)}:i_{R_1}=1,i_{R_{K+1}}=K+1} p_{s_{\mathcal{I}}^{(1)}|\sigma_1} \log \frac{p_{s_{\mathcal{I}}^{(1)}|\sigma_1}}{p_{s_{\mathcal{I}}^{(1)}|\sigma_{K+1}}} + \sum_{R_1<R_{K+1}} \sum_{s_{\mathcal{I}}^{(1)}:i_{R_1}=K+1,i_{R_{K+1}}=1} p_{s_{\mathcal{I}}^{(1)}|\sigma_1} \log \frac{p_{s_{\mathcal{I}}^{(1)}|\sigma_1}}{p_{s_{\mathcal{I}}^{(1)}|\sigma_{K+1}}}$$

$$= \sum_{R_1<R_{K+1}} \sum_{s_{\mathcal{I}}^{(1)}:i_{R_1}=1,i_{R_{K+1}}=K+1} p_{s_{\mathcal{I}}^{(1)}|\sigma_1} \log \frac{p_{s_{\mathcal{I}}^{(1)}|\sigma_1}}{p_{s_{\mathcal{I}}^{(1)}|\sigma_{K+1}}} + \sum_{R_1<R_{K+1}} \sum_{s_{\mathcal{I}}^{(1)}:i_{R_1}=1,i_{R_{K+1}}=K+1} p_{s_{\mathcal{I}}^{(1)}|\sigma_{K+1}} \log \frac{p_{s_{\mathcal{I}}^{(1)}|\sigma_{K+1}}}{p_{s_{\mathcal{I}}^{(1)}|\sigma_1}}$$

$$= \sum_{R_1<R_{K+1}} \sum_{s_{\mathcal{I}}^{(1)}:i_{R_1}=1,i_{R_{K+1}}=K+1} \left( p_{s_{\mathcal{I}}^{(1)}|\sigma_1} - p_{s_{\mathcal{I}}^{(1)}|\sigma_{K+1}} \right) \log \frac{p_{s_{\mathcal{I}}^{(1)}|\sigma_1}}{p_{s_{\mathcal{I}}^{(1)}|\sigma_{K+1}}}. \tag{131}$$

*Case 4* : $1 \notin \mathcal{I}$ and $K+1 \notin \mathcal{I}$:

The scores of items for any given $\mathcal{I}$ are unchanged given either $\sigma = \sigma_1$ or $\sigma = \sigma_{K+1}$. Computing the KL divergence, which we will denote by $D_4$, we get

$$D_4 = D(P_{s_{\mathcal{I}}^{(1)}|\sigma=\sigma_1} || P_{s_{\mathcal{I}}^{(1)}|\sigma=\sigma_{K+1}}) = 0. \tag{132}$$

Putting altogether, we can express $\sum_{\mathcal{I}} D(P_{s_{\mathcal{I}}^{(1)}|\sigma=\sigma_1} || P_{s_{\mathcal{I}}^{(1)}|\sigma=\sigma_{K+1}})$ as follows.

$$\sum_{\mathcal{I}} D(P_{s_{\mathcal{I}}^{(1)}|\sigma=\sigma_1} || P_{s_{\mathcal{I}}^{(1)}|\sigma=\sigma_{K+1}}) = \binom{n-2}{M-1}(D_1 + D_2) + \binom{n-2}{M-2}D_3. \tag{133}$$

Soon, we will show that

$$D_1 + D_2 \le c_0 \Delta_K^2, \tag{134}$$
$$D_3 \le c_0 \Delta_K^2, \tag{135}$$

where $c_0$ is a numerical constant.

Applying (134) and (135) to (133), we get the claimed bound (122):

$$\sum_{\mathcal{I}} D(P_{s_{\mathcal{I}}^{(1)}|\sigma=\sigma_1} || P_{s_{\mathcal{I}}^{(1)}|\sigma=\sigma_{K+1}}) = \binom{n-2}{M-1}(D_1 + D_2) + \binom{n-2}{M-2}D_3 \le \binom{n-2}{M-1}c_0\Delta_K^2 + \binom{n-2}{M-2}c_0\Delta_K^2 \tag{136}$$

$$\le \binom{n}{M}\frac{M(n-M)}{n(n-1)}c_0\Delta_K^2 + \binom{n}{M}\frac{M(M-1)}{n(n-1)}c_0\Delta_K^2 \le \binom{n}{M}\frac{M}{n}c_0\Delta_K^2. \tag{137}$$

Now, let us provide the proofs of (134) and (135).

### 4.1.1 Bound on $D_1 + D_2$: Proof of (134)

From (127) and (128), we get

$$D_1 + D_2 = \sum_{R=1}^{M} \sum_{s_{\mathcal{I}}^{(1)}:i_R=1} \left(p_{s_{\mathcal{I}}^{(1)}|\sigma_1} - p_{s_{\mathcal{I}}^{(1)}|\sigma_{K+1}}\right) \log \frac{p_{s_{\mathcal{I}}^{(1)}|\sigma_1}}{p_{s_{\mathcal{I}}^{(1)}|\sigma_{K+1}}} = \sum_{R=1}^{M} \sum_{s_{\mathcal{I}}^{(1)}:i_R=1} p_{s_{\mathcal{I}}^{(1)}|\sigma_{K+1}} \left(\frac{p_{s_{\mathcal{I}}^{(1)}|\sigma_1}}{p_{s_{\mathcal{I}}^{(1)}|\sigma_{K+1}}} - 1\right) \log \frac{p_{s_{\mathcal{I}}^{(1)}|\sigma_1}}{p_{s_{\mathcal{I}}^{(1)}|\sigma_{K+1}}}. \tag{138}$$

To obtain an upper bound of $D_1 + D_2$, we will show that for $s_{\mathcal{I}}^{(1)} = (i_1, i_2, \ldots, i_R = 1, \ldots, i_M)$, where $R$ is the rank of item 1 within the permutation $s_{\mathcal{I}}^{(1)}$, we have

$$\left(\frac{p_{s_{\mathcal{I}}^{(1)}|\sigma_1}}{p_{s_{\mathcal{I}}^{(1)}|\sigma_{K+1}}} - 1\right) \log \frac{p_{s_{\mathcal{I}}^{(1)}|\sigma_1}}{p_{s_{\mathcal{I}}^{(1)}|\sigma_{K+1}}} \le b^2\Delta_K^2 + b^2\Delta_K^2 \left(\frac{1}{M-R+1} + \log \frac{M}{M-R+1}\right)^2, \tag{139}$$

$$\sum_{s_{\mathcal{I}}^{(1)}:i_R=1} p_{s_{\mathcal{I}}^{(1)}|\sigma_{K+1}} \le \left(3M^{-1/b}\right)\left(\frac{1}{M-R+1}\right)^{1-1/b}, \tag{140}$$

where $b := \frac{w_{\max}}{w_{\min}}$. We provide the proofs of (139) and (140) in detail soon. For the time being, let us show that $D_1 + D_2$ attains the claimed bound of (134).

Applying (139) to (138), we get

$$D_1 + D_2 \le \sum_{R=1}^{M} \sum_{s_{\mathcal{I}}^{(1)}:i_R=1} p_{s_{\mathcal{I}}^{(1)}|\sigma_{K+1}}b^2\Delta_K^2 + \sum_{R=1}^{M} \sum_{s_{\mathcal{I}}^{(1)}:i_R=1} p_{s_{\mathcal{I}}^{(1)}|\sigma_{K+1}}b^2\Delta_K^2 \left(\frac{1}{M-R+1} + \log \frac{M}{M-R+1}\right)^2 \tag{141}$$

$$\stackrel{(a)}{=} b^2\Delta_K^2 + b^2\Delta_K^2 \sum_{R=1}^{M} \sum_{s_{\mathcal{I}}^{(1)}:i_R=1} p_{s_{\mathcal{I}}^{(1)}|\sigma_{K+1}} \left(\frac{1}{M-R+1} + \log \frac{M}{M-R+1}\right)^2 \tag{142}$$

$$= b^2\Delta_K^2 + b^2\Delta_K^2 \sum_{R=1}^{M} \left[\left(\frac{1}{M-R+1} + \log \frac{M}{M-R+1}\right)^2 \sum_{s_{\mathcal{I}}^{(1)}:i_R=1} p_{s_{\mathcal{I}}^{(1)}|\sigma_{K+1}}\right], \tag{143}$$

where $(a)$ follows by the fact that the sum of the probabilities of all possible permutations is 1. Then, applying (140) to (143), we get

$$D_1 + D_2 \leq b^2\Delta_K^2 \left(1 + \left(3M^{-1/b}\right) \sum_{R=1}^{M} \left(\frac{1}{M-R+1}\right)^{1-1/b} \left(\frac{1}{M-R+1} + \log \frac{M}{M-R+1}\right)^2\right) \leq c_0\Delta_K^2, \tag{144}$$

where (144) follows by the fact that $\sum_{R=1}^{M} \left(\frac{1}{M-R+1}\right)^{1-1/b} \left(\frac{1}{M-R+1} + \log \frac{M}{M-R+1}\right)^2 \leq cM^{1/b}$ as shown in Appendix 5.2. This ends the proof of (134).

Now, we provide the proofs of (139) and (140).

**Proof of (139):** For $s_{\mathcal{I}}^{(1)} = (i_1, i_2, \ldots, i_R = 1, \ldots, i_M)$, we can obtain an upper bound of $\left(\frac{p_{s_{\mathcal{I}}^{(1)}|\sigma_1}}{p_{s_{\mathcal{I}}^{(1)}|\sigma_{K+1}}} - 1\right) \log \frac{p_{s_{\mathcal{I}}^{(1)}|\sigma_1}}{p_{s_{\mathcal{I}}^{(1)}|\sigma_{K+1}}}$ as follows.

$$\left(\frac{p_{s_{\mathcal{I}}^{(1)}|\sigma_1}}{p_{s_{\mathcal{I}}^{(1)}|\sigma_{K+1}}} - 1\right) \log \frac{p_{s_{\mathcal{I}}^{(1)}|\sigma_1}}{p_{s_{\mathcal{I}}^{(1)}|\sigma_{K+1}}} = \left(\frac{w_K}{w_{K+1}} \prod_{r=1}^{R} \frac{w_{K+1} + \sum_{m:m\in[r,M]\backslash\{R\}} w_{i_m}}{w_K + \sum_{m:m\in[r,M]\backslash\{R\}} w_{i_m}} - 1\right) \log \left(\frac{w_K}{w_{K+1}} \prod_{r=1}^{R} \frac{w_{K+1} + \sum_{m:m\in[r,M]\backslash\{R\}} w_{i_m}}{w_K + \sum_{m:m\in[r,M]\backslash\{R\}} w_{i_m}}\right) \tag{145}$$

$$= \left(\frac{w_K}{w_{K+1}} \prod_{r=1}^{R} \frac{w_{K+1} + \sum_{m:m\in[r,M]\backslash\{R\}} w_{i_m}}{w_K + \sum_{m:m\in[r,M]\backslash\{R\}} w_{i_m}} - 1\right) \log \left(\frac{w_K}{w_{K+1}}\right) \tag{146}$$

$$+ \left(1 - \frac{w_K}{w_{K+1}} \prod_{r=1}^{R} \frac{w_{K+1} + \sum_{m:m\in[r,M]\backslash\{R\}} w_{i_m}}{w_K + \sum_{m:m\in[r,M]\backslash\{R\}} w_{i_m}}\right) \log \left(\prod_{r=1}^{R} \frac{w_K + \sum_{m:m\in[r,M]\backslash\{R\}} w_{i_m}}{w_{K+1} + \sum_{m:m\in[r,M]\backslash\{R\}} w_{i_m}}\right) \tag{147}$$

$$\stackrel{(a)}{<} \left(\frac{w_K - w_{K+1}}{w_{K+1}}\right) \log \left(\frac{w_K}{w_{K+1}}\right) + \left(1 - \frac{1}{\prod_{r=1}^{R}\left(1 + \frac{w_K - w_{K+1}}{w_{K+1} + \sum_{m:m\in[r,M]\backslash\{R\}} w_{i_m}}\right)}\right) \log \prod_{r=1}^{R} \left(1 + \frac{w_K - w_{K+1}}{w_{K+1} + \sum_{m:m\in[r,M]\backslash\{R\}} w_{i_m}}\right) \tag{148}$$

$$\stackrel{(b)}{\leq} \left(\frac{w_K - w_{K+1}}{w_{K+1}}\right) \log \left(\frac{w_K}{w_{K+1}}\right) + \left(1 - \frac{1}{\prod_{r=1}^{R}\left(1 + \frac{w_K - w_{K+1}}{(M-r+1)w_{K+1}}\right)}\right) \log \prod_{r=1}^{R} \left(1 + \frac{w_K - w_{K+1}}{(M-r+1)w_{K+1}}\right) \tag{149}$$

$$\stackrel{(c)}{\leq} \left(\frac{w_K - w_{K+1}}{w_{K+1}}\right) \log \left(1 + \frac{w_K - w_{K+1}}{w_{K+1}}\right) + \left(\sum_{r=1}^{R} \log \left(1 + \frac{w_K - w_{K+1}}{(M-r+1)w_{K+1}}\right)\right)^2 \tag{150}$$

$$\stackrel{(d)}{\leq} \left(\frac{w_K - w_{K+1}}{w_{K+1}}\right)^2 + \left(\frac{w_K - w_{K+1}}{w_{K+1}} \sum_{r=1}^{R} \frac{1}{M-r+1}\right)^2 \tag{151}$$

$$\stackrel{(e)}{\leq} \left(\frac{w_K - w_{K+1}}{w_{K+1}}\right)^2 \left[1 + \left(\frac{1}{M-R+1} + \log \frac{M}{M-R+1}\right)^2\right] \stackrel{(f)}{\leq} b^2\Delta_K^2 \left[1 + \left(\frac{1}{M-R+1} + \log \frac{M}{M-R+1}\right)^2\right], \tag{152}$$

where $(a)$ follows by the fact that $\prod_{r=1}^{R} \frac{w_{K+1} + \sum_{m=r}^{R-1} w_{i_m} + \sum_{m=R+1}^{M} w_{i_m}}{w_K + \sum_{m=r}^{R-1} w_{i_m} + \sum_{m=R+1}^{M} w_{i_m}} < 1$ since $w_K > w_{K+1}$; $(b)$ follows by replacing $w_{i_m}$ with $w_{K+1}$; $(c)$ follows by applying $1 - e^{-x} \leq x$; $(d)$ follows by applying $\log(1+x) \leq x$; $(e)$ follows by the

fact that $\sum_{r=1}^{R} \frac{1}{M-r+1} \leq \frac{1}{M-R+1} + \log \frac{M}{M-R+1}$ as shown in Appendix 5.1; $(f)$ follows by $b = \frac{w_{\max}}{w_{\min}} = \frac{w_K}{w_{K+1}}$ and $\Delta_K = \frac{w_K - w_{K+1}}{w_{\max}} = \frac{w_K - w_{K+1}}{w_K}$.

**Proof of (140):** Now, we provide an upper bound of $p_{s_{\mathcal{I}}^{(1)}|\sigma_{K+1}}$.

$$\sum_{s_{\mathcal{I}}^{(1)}:i_R=1} p_{s_{\mathcal{I}}^{(1)}|\sigma_{K+1}} = \sum_{s_{\mathcal{I}}^{(1)}:i_R=1} \left( \prod_{r=1}^{R-1} \frac{w_{i_r}}{w_{K+1} + \sum_{m:m\in[r,M]\setminus\{R\}} w_{i_m}} \right) \left( \prod_{r=R+1}^{M} \frac{w_{i_r}}{\sum_{m=r}^{M} w_{i_m}} \right) \frac{w_{K+1}}{w_{K+1} + \sum_{m=R+1}^{M} w_{i_m}}$$
(153)

$$\overset{(a)}{=} \sum_{s_{\mathcal{I}}^{(1)}:i_R=1} \left( \prod_{r=1}^{R-1} \frac{w_{i_r}}{\sum_{m:m\in[r,M]\setminus\{R\}} w_{i_m}} \right) \left( \prod_{r=1}^{R-1} \frac{\sum_{m:m\in[r,M]\setminus\{R\}} w_{i_m}}{w_{K+1} + \sum_{m:m\in[r,M]\setminus\{R\}} w_{i_m}} \right) \left( \prod_{r=R+1}^{M} \frac{w_{i_r}}{\sum_{m=r}^{M} w_{i_m}} \right) \frac{w_{K+1}}{w_{K+1} + \sum_{m=R+1}^{M} w_{i_m}}$$
(154)

$$= \sum_{s_{\mathcal{I}}^{(1)}:i_R=1} \left( \prod_{r=1}^{R-1} \frac{w_{i_r}}{\sum_{m:m\in[r,M]\setminus\{R\}} w_{i_m}} \right) \left( \prod_{r=R+1}^{M} \frac{w_{i_r}}{\sum_{m=r}^{M} w_{i_m}} \right) \left( \prod_{r=1}^{R-1} \frac{\sum_{m:m\in[r,M]\setminus\{R\}} w_{i_m}}{w_{K+1} + \sum_{m:m\in[r,M]\setminus\{R\}} w_{i_m}} \right) \frac{w_{K+1}}{w_{K+1} + \sum_{m=R+1}^{M} w_{i_m}}$$
(155)

$$\overset{(b)}{\leq} \sum_{s_{\mathcal{I}}^{(1)}:i_R=1} \left( \prod_{r=1}^{R-1} \frac{w_{i_r}}{\sum_{m:m\in[r,M]\setminus\{R\}} w_{i_m}} \right) \left( \prod_{r=R+1}^{M} \frac{w_{i_r}}{\sum_{m=r}^{M} w_{i_m}} \right) \left( \prod_{r=1}^{R-1} \frac{(M-r)w_K}{w_{K+1} + (M-r)w_K} \right) \frac{w_{K+1}}{(M-R+1)w_{K+1}}$$
(156)

$$\overset{(c)}{=} \left( \prod_{r=1}^{R-1} \frac{(M-r)w_K}{w_{K+1} + (M-r)w_K} \right) \frac{w_{K+1}}{(M-R+1)w_{K+1}} = \left( \prod_{r=1}^{R-1} \frac{w_K + (M-r)w_K}{w_{K+1} + (M-r)w_K} \right) \frac{1}{M} \leq \left( \prod_{r=1}^{R-1} 1 + \frac{w_K - w_{K+1}}{(M-r)w_K} \right) \frac{1}{M}$$
(157)

$$\overset{(d)}{\leq} \exp \left( \frac{w_K - w_{K+1}}{w_K} \sum_{r=1}^{R-1} \frac{1}{M-r} \right) \frac{1}{M} \overset{(e)}{\leq} \exp \left( \frac{w_K - w_{K+1}}{w_K} \left( \frac{1}{M-R+1} + \log \frac{M-1}{M-R+1} \right) \right) \frac{1}{M}$$
(158)

$$\overset{(f)}{\leq} 3 \left( \frac{M-1}{M-R+1} \right)^{\frac{w_K - w_{K+1}}{w_K}} \frac{1}{M} \overset{(g)}{\leq} \left( 3M^{-1/b} \right) \left( \frac{1}{M-R+1} \right)^{1-1/b},$$
(159)

where $(a)$ follows by splitting the first product in parentheses into two; $(b)$ follows by replacing $w_{i_m}$ with $w_K$ or $w_{K+1}$ properly; $(c)$ follows by the fact that $\sum_{s_{\mathcal{I}}^{(1)}:i_R=1} \left( \prod_{r=1}^{R-1} \frac{w_{i_r}}{\sum_{m:m\in[r,M]\setminus\{R\}} w_{i_m}} \right) \left( \prod_{r=R+1}^{M} \frac{w_{i_r}}{\sum_{m=r}^{M} w_{i_m}} \right) = 1$ holds according to the PL model; $(d)$ follows by applying $1 + x \leq e^x$; $(e)$ follows by the fact that $\sum_{r=1}^{R-1} \frac{1}{M-r} \leq \frac{1}{M-R+1} + \log \frac{M-1}{M-R+1}$ as shown in Appendix 5.1; $(f)$ follows by the fact that $\exp \left( \frac{w_K - w_{K+1}}{w_K} \frac{1}{M-R+1} \right) \leq e \leq 3$ for any $R$; $(g)$ follows by the fact that $\frac{w_{K+1}}{w_K} \leq \frac{1}{b}$. Note that $b = \Theta(1)$.

### 4.1.2   Bound on $D_3$: Proof of (135)

Following a similar line of steps toward the bound of $D_1 + D_2$, we can also obtain an upper bound of $D_3$. From (131), we get

$$D_3 = \sum_{R_1 < R_{K+1}} \sum_{s_{\mathcal{I}}^{(1)}:i_{R_1}=1, i_{R_{K+1}}=K+1} \left( p_{s_{\mathcal{I}}^{(1)}|\sigma_1} - p_{s_{\mathcal{I}}^{(1)}|\sigma_{K+1}} \right) \log \frac{p_{s_{\mathcal{I}}^{(1)}|\sigma_1}}{p_{s_{\mathcal{I}}^{(1)}|\sigma_{K+1}}}$$
(160)

$$= \sum_{R_1 < R_{K+1}} \sum_{s_{\mathcal{I}}^{(1)}:i_{R_1}=1, i_{R_{K+1}}=K+1} p_{s_{\mathcal{I}}^{(1)}|\sigma_{K+1}} \left( \frac{p_{s_{\mathcal{I}}^{(1)}|\sigma_1}}{p_{s_{\mathcal{I}}^{(1)}|\sigma_{K+1}}} - 1 \right) \log \frac{p_{s_{\mathcal{I}}^{(1)}|\sigma_1}}{p_{s_{\mathcal{I}}^{(1)}|\sigma_{K+1}}}.$$
(161)

Similarly, we will show that for $s_{\mathcal{I}}^{(1)} = (i_1, i_2, \ldots, i_{R_1} = 1, \ldots, i_{R_{K+1}} = K+1, \ldots, i_M)$, where $R_1$ is the rank of item

1 and $R_{K+1}$ is the rank of item $K+1$ respectively within the permutation $s_{\mathcal{I}}^{(1)}$, we have

$$\left(\frac{p_{s_{\mathcal{I}}^{(1)}|\sigma_1}}{p_{s_{\mathcal{I}}^{(1)}|\sigma_{K+1}}} - 1\right)\log\frac{p_{s_{\mathcal{I}}^{(1)}|\sigma_1}}{p_{s_{\mathcal{I}}^{(1)}|\sigma_{K+1}}} \le b^2\Delta_K^2\left(\frac{1}{M-R_{K+1}+1} + \log\frac{M}{M-R_{K+1}+1}\right)^2, \tag{162}$$

$$\sum_{s_{\mathcal{I}}^{(1)}:i_{R_1}=1,i_{R_{K+1}}=K+1} p_{s_{\mathcal{I}}^{(1)}|\sigma_{K+1}} \le \frac{b}{M-1}\left(3M^{-1/b}\right)\left(\frac{1}{M-R_{K+1}+1}\right)^{1-1/b}. \tag{163}$$

Applying (162) and (163) to (161), we get

$$D_3 \overset{(a)}{\le} \sum_{R_1<R_{K+1}}\sum_{s_{\mathcal{I}}^{(1)}:i_{R_1}=1,i_{R_{K+1}}=K+1} p_{s_{\mathcal{I}}^{(1)}|\sigma_{K+1}}b^2\Delta_K^2\left(\frac{1}{M-R_{K+1}+1}+\log\frac{M}{M-R_{K+1}+1}\right)^2 \tag{164}$$

$$= \sum_{R_1<R_{K+1}} b^2\Delta_K^2\left(\frac{1}{M-R_{K+1}+1}+\log\frac{M}{M-R_{K+1}+1}\right)^2 \sum_{s_{\mathcal{I}}^{(1)}:i_{R_1}=1,i_{R_{K+1}}=K+1} p_{s_{\mathcal{I}}^{(1)}|\sigma_{K+1}} \tag{165}$$

$$\overset{(b)}{=} \sum_{R_1<R_{K+1}} \frac{b}{M-1}\left(3M^{-1/b}\right)\left(\frac{1}{M-R_{K+1}+1}\right)^{1-1/b} b^2\Delta_K^2\left(\frac{1}{M-R_{K+1}+1}+\log\frac{M}{M-R_{K+1}+1}\right)^2 \tag{166}$$

$$\overset{(c)}{\le} \sum_{R_{K+1}=1}^{M} (M-1)\frac{b}{M-1}\left(3M^{-1/b}\right)\left(\frac{1}{M-R_{K+1}+1}\right)^{1-1/b} b^2\Delta_K^2\left(\frac{1}{M-R_{K+1}+1}+\log\frac{M}{M-R_{K+1}+1}\right)^2 \tag{167}$$

$$\le 3b^3\Delta_K^2 M^{-1/b}\sum_{R_{K+1}=1}^{M}\left(\frac{1}{M-R_{K+1}+1}\right)^{1-1/b}\left(\frac{1}{M-R_{K+1}+1}+\log\frac{M}{M-R_{K+1}+1}\right)^2 \overset{(d)}{\le} c_0\Delta_K^2, \tag{168}$$

where $(a)$ follows by (162); $(b)$ follows by (163); $(c)$ follows by the fact that we can have at most $(M-1)$ different pairs of $(R_1, R_{K+1})$ for fixed $R_{K+1}$; $(d)$ follows by the fact that $\sum_{R_{K+1}=1}^{M}\left(\frac{1}{M-R_{K+1}+1}\right)^{1-1/b}\left(\frac{1}{M-R_{K+1}+1}+\log\frac{M}{M-R_{K+1}+1}\right)^2 \le cM^{1/b}$ as shown in Appendix 5.2. This ends the proof of (135).

Now, we provide proofs of (162) and (163).

**Proof of (162):** We can obtain an upper bound of $\left(\frac{p_{s_{\mathcal{I}}^{(1)}|\sigma_1}}{p_{s_{\mathcal{I}}^{(1)}|\sigma_{K+1}}} - 1\right)\log\frac{p_{s_{\mathcal{I}}^{(1)}|\sigma_1}}{p_{s_{\mathcal{I}}^{(1)}|\sigma_{K+1}}}$ by applying a similar line of steps from (148) to (152).

$$\left(\frac{p_{s_{\mathcal{I}}^{(1)}|\sigma_1}}{p_{s_{\mathcal{I}}^{(1)}|\sigma_{K+1}}} - 1\right)\log\frac{p_{s_{\mathcal{I}}^{(1)}|\sigma_1}}{p_{s_{\mathcal{I}}^{(1)}|\sigma_{K+1}}} = \left(1 - \left(\prod_{r=R_1+1}^{R_{K+1}}\frac{w_K+\sum_{m:m\in[r,M]\setminus\{R\}}w_{i_m}}{w_{K+1}+\sum_{m:m\in[r,M]\setminus\{R\}}w_{i_m}}\right)^{-1}\right)\log\prod_{r=R_1+1}^{R_{K+1}}\frac{w_K+\sum_{m:m\in[r,M]\setminus\{R\}}w_{i_m}}{w_{K+1}+\sum_{m:m\in[r,M]\setminus\{R\}}w_{i_m}} \tag{169}$$

$$\le b^2\Delta_K^2\left(\frac{1}{M-R_{K+1}+1} + \log\frac{M}{M-R_{K+1}+1}\right)^2. \tag{170}$$

**Proof of (163):** Now, we provide the upper bound of $\sum_{s_{\mathcal{I}}^{(1)}:i_{R_1}=1,i_{R_{K+1}}=K+1} p_{s_{\mathcal{I}}^{(1)}|\sigma_{K+1}}$.

$$\sum_{s_{\mathcal{I}}^{(1)}:i_{R_1}=1,i_{R_{K+1}}=K+1} p_{s_{\mathcal{I}}^{(1)}|\sigma_{K+1}}$$

$$\overset{(a)}{=} \sum_{s_{\mathcal{I}}^{(1)}:i_{R_1}=1,i_{R_{K+1}}=K+1}\left(\prod_{r=1}^{R_1-1}\frac{w_{i_r}}{\sum_{m:m\in[r,M]\setminus\{R_1,R_{K+1}\}}w_{i_m}}\right)\left(\prod_{r=R_1+1}^{R_{K+1}-1}\frac{w_{i_r}}{\sum_{m:m\in[r,M]\setminus\{R_{K+1}\}}w_{i_m}}\right)\left(\prod_{r=R_{K+1}+1}^{M}\frac{w_{i_r}}{\sum_{m=r}^{M}w_{i_m}}\right)$$

$$\times \left( \frac{w_{K+1}}{w_{K+1} + w_K + \sum_{m:m\in[R_1,M]\setminus\{R_1,R_{K+1}\}} w_{i_m}} \right) \left( \frac{w_K}{w_K + \sum_{m=R_{K+1}+1}^{M} w_{i_m}} \right)$$

$$\times \left( \prod_{r=1}^{R_1-1} \frac{\sum_{m:m\in[r,M]\setminus\{R_1,R_{K+1}\}} w_{i_m}}{w_{K+1} + w_K + \sum_{m:m\in[r,M]\setminus\{R_1,R_{K+1}\}} w_{i_m}} \right) \left( \prod_{r=R_1+1}^{R_{K+1}-1} \frac{\sum_{m:m\in[r,M]\setminus\{R_{K+1}\}} w_{i_m}}{w_K + \sum_{m:m\in[r,M]\setminus\{R_{K+1}\}} w_{i_m}} \right) \tag{171}$$

$$\overset{(b)}{\leq} \left( \frac{w_{K+1}}{(M-R_1+1)w_{K+1}} \right) \left( \frac{w_K}{(M-R_{K+1}+1)w_{K+1}} \right) \left( \prod_{r=1}^{R_1-1} \frac{(M-r)w_K}{w_{K+1}+(M-r)w_K} \right) \left( \prod_{r=R_1+1}^{R_{K+1}-1} \frac{(M-r)w_K}{(M-r+1)w_K} \right) \tag{172}$$

$$\times \left[ \sum_{s_{\mathcal{I}}^{(1)}:i_{R_1}=1,i_{R_{K+1}}=K+1} \left( \prod_{r=1}^{R_1-1} \frac{w_{i_r}}{\sum_{m:m\in[r,M]\setminus\{R_1,R_{K+1}\}} w_{i_m}} \right) \left( \prod_{r=R_1+1}^{R_{K+1}-1} \frac{w_{i_r}}{\sum_{m:m\in[r,M]\setminus\{R_{K+1}\}} w_{i_m}} \right) \left( \prod_{r=R_{K+1}+1}^{M} \frac{w_{i_r}}{\sum_{m=r}^{M} w_{i_m}} \right) \right] \tag{173}$$

$$\overset{(c)}{=} \left( \frac{w_{K+1}}{(M-R_1+1)w_{K+1}} \right) \left( \frac{w_K}{(M-R_{K+1}+1)w_{K+1}} \right) \left( \prod_{r=1}^{R_1-1} \frac{(M-r)w_K}{w_{K+1}+(M-r)w_K} \right) \left( \prod_{r=R_1+1}^{R_{K+1}-1} \frac{(M-r)w_K}{(M-r+1)w_K} \right) \tag{174}$$

$$\overset{(d)}{\leq} \frac{b}{M-1} \prod_{r=1}^{R_1-1} \left( 1 + \frac{w_K - w_{K+1}}{w_{K+1}+(M-r)w_K} \right) \frac{1}{M} \leq \frac{b}{M-1} \prod_{r=1}^{R_1-1} \left( 1 + \frac{w_K - w_{K+1}}{(M-r)w_K} \right) \frac{1}{M} \tag{175}$$

$$\overset{(e)}{\leq} \frac{b}{M-1} \prod_{r=1}^{R_{K+1}-1} \left( 1 + \frac{w_K - w_{K+1}}{(M-r)w_K} \right) \frac{1}{M} \overset{(g)}{\leq} \frac{b}{M-1} \left( 3M^{-1/b} \right) \left( \frac{1}{M-R_{K+1}+1} \right)^{1-1/b}, \tag{176}$$

where $(a)$ follows by applying a similar line of step from (153) to (155); $(b)$ follows by replacing $w_{i_m}$ with $w_K$ or $w_{K+1}$ properly; $(c)$ follows by the fact that the term in brackets is equal to 1 according to the PL model; $(e)$ follows by applying a similar line of steps in(157); $(d)$ follows by the fact that $R_1 < R_{K+1}$; $(e)$ follows by applying a similar line of steps from (158) to (159).

# 5 Appendix

## 5.1 Bound of Summation 1

For all $\alpha, \beta > 0$,

$$\sum_{m=\alpha}^{\beta} \frac{1}{m} \leq \frac{1}{\alpha} + \log \frac{\beta}{\alpha}. \tag{177}$$

*Proof.* $\frac{1}{m} \leq \int_{m-1}^{m} \frac{1}{x} dx = \log m - \log(m-1)$ holds since $\frac{1}{x}$ is a decreasing function for $x \in (0,\infty)$. $\square$

## 5.2 Bound of Summation 2

For all $\gamma$ such that $0 < \gamma \leq 1$ and $\gamma = \Theta(1)$,

$$\sum_{R=1}^{M} \left( \frac{1}{M-R+1} \right)^{1-\gamma} \left( \frac{1}{M-R+1} + \log \frac{M}{M-R+1} \right)^2 \leq cM^\gamma, \tag{178}$$

where $c$ is a numerical constant.

*Proof.* We can rewrite (178) as

$$\sum_{m=1}^{M} \left(\frac{1}{m}\right)^{1-\gamma} \left(\frac{1}{m} + \log \frac{M}{m}\right)^2 \le cM^\gamma. \tag{179}$$

Since $\left(\frac{1}{x}\right)^{1-\gamma} \left(\frac{1}{x} + \log \frac{M}{x}\right)^2$ is a decreasing function for $x \in [1, M]$, we can see that

$$\left(\frac{1}{m}\right)^{1-\gamma} \left(\frac{1}{m} + \log \frac{M}{m}\right)^2 \le \int_{m-1}^{m} \left(\frac{1}{x}\right)^{1-\gamma} \left(\frac{1}{x} + \log \frac{M}{x}\right)^2 dx. \tag{180}$$

Thus, for $0 < \gamma < 1$, we get

$$\sum_{m=1}^{M} \left(\frac{1}{m}\right)^{1-\gamma} \left(\frac{1}{m} + \log \frac{M}{m}\right)^2 \le (1 + \log M)^2 + \int_{1}^{M} \left(\frac{1}{x}\right)^{1-\gamma} \left(\frac{1}{x} + \log \frac{M}{x}\right)^2 dx \tag{181}$$

$$\le (1 + \log M)^2 + M^\gamma \left(\frac{1}{M^2(-2+\gamma)} + \frac{2}{M(-1+\gamma)^2} + \frac{2}{\gamma^3}\right) + c\frac{(\log M)^2}{\gamma} \le c'M^\gamma, \tag{182}$$

where $c$ and $c'$ are some numerical constants. For $\gamma = 1$, we get

$$\sum_{m=1}^{M} \left(\frac{1}{m} + \log \frac{M}{m}\right)^2 \le (1 + \log M)^2 + \int_{1}^{M} \left(\frac{1}{x} + \log \frac{M}{x}\right)^2 dx = 2M + (\log M)^2 - \frac{1}{M} \le 3M. \tag{183}$$

$\square$

## 5.3   Hoeffding's Inequality

Throughout the provided proofs, we often use Hoeffding's inequality stated as follows.
**Hoeffding's inequality** *Let $X_i$ be independent random variables bounded by the interval $[a_i, b_i] : a_i \le X_i \le b_i$. Then,*

$$\Pr\left[\left|\sum_{i=1}^{n} X_i - \sum_{i=1}^{n} \mathbb{E}[X_i]\right| \ge t\right] \le 2\exp\left(-\frac{2t^2}{\sum_{i=1}^{n}(b_i - a_i)^2}\right). \tag{184}$$

## 5.4   Hölder's Inequality

Throughout the provided proofs, we often use the matrix version of Hölder's inequality stated as follows.
**Hölder's inequality** *For a matrix $Q \in \mathbb{R}^{m \times n}$, the following inequality holds.*

$$\|Q\|_2 \le \sqrt{\|Q\|_1 \|Q\|_\infty}. \tag{185}$$

Using the definitions of $\|Q\|_1$ and $\|Q\|_\infty$, we can further derive that

$$\|Q\|_2 \le \sqrt{\left(\max_j \sum_{i=1}^{m} |Q_{ij}|\right)\left(\max_i \sum_{j=1}^{n} |Q_{ij}|\right)}. \tag{186}$$

## 5.5 Matrix Bernstein Inequality

Throughout the provided proofs, we often use the matrix Bernstein inequality stated as follows.

**Matrix Bernstein Inequality (Tropp 2011)** *Consider a finite sequence $\{Q_i\}$ of independent, random, self-adjoint $n \times n$ matrices. Assume that*

$$\mathbb{E}[Q_i] = 0 \quad and \quad \|Q_i\|_2 \leq R \quad almost\ surely. \tag{187}$$

*Then, for all $t \geq 0$,*

$$\Pr\left[\left\|\sum_i Q_i\right\|_2 \geq t\right] \leq 2n \exp\left(\frac{-t^2/2}{\sigma^2 + Rt}\right), \tag{188}$$

*where $\sigma^2 := \left\|\sum_i \mathbb{E}\left[Q_i^2\right]\right\|_2$.*

## 5.6 Courant-Fischer Theorem

Throughout the provided proofs, we often use a special case of the Courant-Fischer theorem stated as follows.

**Courant-Fischer Theorem** *Consider a symmetric matrix $Q \in \mathbb{R}^{n \times n}$ and its eigenvalue $\{\lambda_i\}_{i=1}^n$. Then,*

$$\max_i |\lambda_i| = \max_{\|x\|=1} \left|x^T Q x\right|. \tag{189}$$

One can show this by representing $x$ using the eigenvectors of $Q$. From the definition of $\|Q\|_2$, we can further derive that

$$\|Q\|_2 = \max_i |\lambda_i| = \max_{\|x\|=1} \left|x^T Q x\right|. \tag{190}$$

## 5.7 Concentration Inequality

Throughout the provided proofs, we often use the a concentration inequality stated as follows.

**Lemma 7.** *Suppose independent and identically distributed (i.i.d.) random variables $X_i$ follow Bernoulli(q) and $q > c\frac{\log n}{n}$. Then, with probability at least $1 - 2n^{-\frac{3r^2}{2(r+3)}c}$,*

$$(1-r)nq \leq \sum_{i=1}^n X_i \leq (1+r)nq. \tag{191}$$

*Proof.* Applying the Bernstein inequality, we get

$$\mathbb{P}\left[\left|\sum_{i=1}^n X_i - nq\right| > t\right] \leq 2\exp\left(-\frac{\frac{1}{2}t^2}{nq + \frac{1}{3}t}\right). \tag{192}$$

Then we choose $t = rnq$ and use $q > c\frac{\log n}{n}$, to get the following tail probability, which completes the proof.

$$\mathbb{P}\left[\left|\sum_{i=1}^n X_i - nq\right| > rnq\right] \leq 2n^{-\frac{3r^2}{2(r+3)}\frac{nq}{\log n}} < 2n^{-\frac{3r^2}{2(r+3)}c}. \tag{193}$$

$\square$

We can see that a sum of random variables concentrates to the order of its expectation with high probability when $c$ and $r$ are constant.