[Reviews · NeurIPS 2017]

Reviewer 1



Paper 1058 Inspired by the pairwise framework for top k ranking aggregation in [17], this paper mainly explores the optimal sample complexity for M-wise top-K rank aggregation in a minmax sense based on the PL-model. To reach that complexity, the authors first derives a lower bound of O(nlgn/(delta_K^2*M)) and then construct a slightly modified Rank Centrality algorithm, which is upper bounded by exactly the same order. Even more surprisingly, unlike [17], such a result is guaranteed without employing the MLE method at all, which is a great improvement of efficiency. The experiments for synthetic datasets show that the performance of Rank Centrality could meet that of Spectral MLE under dense regime, and become slightly worse than Spectral MLE under sparse regime. However, the slight disadvantage of Rank Centrality under sparse regime is not of a big issue considering the efficiency gain. Moreover, the curving fitting of sample complexity perfectly matches $1/M$, the rule suggested by the theoretical results. On the other hand, the experiments for the real world simulated data also suggest the possibility to generalize the main results of this paper to more complicated applications where the PL model assumption doesn’t necessarily hold. Overall, it’s a good paper featured with elegant theoretical analysis and solid experimental supports. It’s an amazing reading experience for me! Thank you so much for sharing your idea and your contributions to this area! There’s only one problem from me: Seeing that the complexity is simultaneously governed by n $delta_K$ and M, what’s the influence of n and $delta_K$ on the practical sample complexity besides M?

Reviewer 2



Summary: The authors consider the problem of optimal top-K rank aggregation from many samples each containing M-wise comparisons. The setup is as follows: There are n items and each has a hidden utility w_i such that w_i >= w_{i+1}. w_K-w_{K+1} has a large separation Delta_K. The task is to identify the top K items. There is a series of samples that are given. Each samples is a preference ordering among M items out of the n chosen. Given the M items in the sample, the preference ordering follows the PL model. Now there is a random hyper graph where each set of M vertices is connected independently with probability p. Further, every hyperedge gives L M-wise i.i.d preference order where M items come from the hyperedge. The authors show that under some density criterion on p, the necessary and sufficient number of samples nis O( n log n/M Delta_K^2 ) . The notable point is that it is inversely proportional to M. The algorithm has two parts: For every M-wise ordered sample, one creates M pariwise orderings that respect the M-wise ordering such that these pairwise orderings are not correlated. This is called sample breaking. Then the problem uses standard algorithmic techniques like Rank Centrality that use the generated pairwise ordered samples to actually generate a top K ranking. The proofs involve justifying the sample breaking step and new analysis is needed to prove the sample complexity results. Strengths: a) For the dense hypergraph case, the work provides sample optimal strategy for aggregation which is the strength of the paper. b) Analyzing rank centrality along with path breaking seems to require a strong l_infinity bound on the entries of the stationary distribution of a Markov Chain whose transition matrix entries have been estimated upto to some additive error. Previous works have provided only l_2 error bounds. This key step helps them separate the top K items from the rest due to the separation assumed. I did not have the time to go through the whole appendix. However I only quickly checked the proof of Theorem 3 which seems to be quite crucial. It seems to be correct. Weaknesses: a) Spectral MLE could be used once sample breaking is employed and new pairwise ordered samples generated. Why has that not been compared with the proposed approach for the experiments in section 4.2 ? Infact that could have been compared even for synthetic experiments in Section 4.1 for M > 2 ? b) The current method's density requirements are not optimal (at least w.r.t the case for M=2). I was wondering if experimentally, sample breaking + some other algorithmic approach can be shown to be better ?

Reviewer 3



This paper considers the problem of performing top-k ranking when provided with M-wise data assuming the Plackett-Luce (PL) model. They provide necessary and sufficient conditions on the sample complexity. The necessary condition employs Han and Verdu's generalized Fano's inequality. For the sufficient condition, the authors give a simple algorithm that essentially "breaks" the M-wise sample to produce pairwise samples, and then uses these in a rank-centrality algorithm. The analysis that this algorithm has the right sample complexity is where the major work of this paper is. The main part of the paper is well written, however, the organization of the supplementary material can be improved. I appreciate the authors clearly indicating the dependencies of their results, but the organization still seems quite fractured. In all, the proof sketch is clearly presented in the text and the discussions are very helpful. This is a solid paper, and I recommend its acceptance.